# Ferritin in Acute Myeloid Leukemia: Not Only a Marker of Inflammation and Iron Overload, but Also a Regulator of Cellular Iron Metabolism, Signaling and Communication

**DOI:** 10.3390/ijms26125744

**Published:** 2025-06-15

**Authors:** Håkon Reikvam, Magnus Gramstad Rolfsnes, Linn Rolsdorph, Miriam Sandnes, Frode Selheim, Maria Hernandez-Valladares, Øystein Bruserud

**Affiliations:** 1Acute Leukemia Research Group, Department of Clinical Science, University of Bergen, 5021 Bergen, Norway; hakon.reikvam@uib.no (H.R.); magnus.rolfsnes@student.uib.no (M.G.R.); miriam.sandnes@helse-bergen.no (M.S.); 2Section for Hematology, Department of Medicine, Haukeland University Hospital, 5021 Bergen, Norway; 3Department of Medicine, Akershus University Hospital, Postboks 1000, 1478 Lørenskog, Norway; linn.rolsdorph@gmail.com; 4Proteomics Unit of University of Bergen (PROBE), University of Bergen, Jonas Lies vei 91, 5009 Bergen, Norway; frode.selheim@uib.no; 5Department of Physical Chemistry, University of Granada, Avenida de la Fuente Nueva S/N, 18071 Granada, Spain; mariahv@ugr.es; 6Instituto de Investigación Biosanitaria ibs.GRANADA, 18012 Granada, Spain

**Keywords:** ferritin, iron, acute myeloid leukemia, acute-phase reaction, ferroptosis, chemotherapy, stem cell transplantation, prognosis

## Abstract

Ferritin is important for cellular iron storage and metabolism. It consists of 24 ferritin heavy- or light-chain subunits surrounding an iron-containing core, but it is also released as an extracellular molecule that shows increased systemic levels during acute-phase reactions. Furthermore, acute myeloid leukemia (AML) is an aggressive bone marrow malignancy that can be associated with increased ferritin levels both at the time of first diagnosis but also during/following anti-AML treatment due to an iron overload. Such high systemic ferritin levels at diagnosis or later allogeneic stem cell transplantation are associated with decreased long-term survival. Extracellular ferritin binds to several receptors expressed by AML cells (e.g., the transferrin receptor and CXCR4 chemokine receptor) and AML-supporting non-leukemic bone marrow cells (e.g., endothelial, mesenchymal or immunocompetent cells). Ferritin can thereby affect the AML cells directly as well as indirectly via AML-supporting neighboring cells. Finally, ferritin should be regarded as a regulator of the dysfunctional iron metabolism that causes increased iron levels in AML cells, and it is important for cell survival through its function during the initial steps of ferroptosis. Thus, ferritin is not only an adverse prognostic biomarker, but also an important regulator of AML cell proliferation, survival and chemosensitivity and the targeting of iron metabolism/ferroptosis is, therefore, a possible strategy in AML therapy.

## 1. Introduction

Acute myeloid leukemia (AML) is an aggressive and heterogeneous malignancy characterized by the proliferation of transformed immature myeloid cells in the bone marrow; the leukemic cells are often detected also in peripheral blood whereas infiltration of other organs is uncommon [1]. Early experimental studies suggest that iron/transferrin enhances the in vitro proliferation of primary AML cells [2,3]. To the best of our knowledge the first study to suggest an association between iron metabolism and AML cell chemosensitivity described an association between decreased AML-free survival and an AML cell phenotype characterized by the altered mRNA expression of several regulators of the cellular iron metabolism (including decreased levels of several vacuolar ATPase components). This adverse AML cell phenotype was also characterized by a generally low capacity of a constitutive cytokine release (i.e., altered cellular communication) [4].

Ferritin is a protein complex consisting of 24 subunits that surround an iron-containing core [5,6]. This protein is responsible for intracellular iron storage, but is also important for the regulation of the cellular iron metabolism as well as the intracellular free iron concentration [7], and it is also involved in cellular protection against oxidative stress [8]. Furthermore, ferritin is a part of the acute-phase reaction and increased serum/plasma levels can be detected during infection/inflammation [9]. Finally, ferritin has immunoregulatory functions that are probably mediated by the ligation of specific receptors expressed by immunocompetent cells [10]. In this review, we discuss the possible importance of ferritin in non-APL (acute promyelocytic leukemia) variants of AML:In the first part, we briefly describe the most important structural and functional characteristics of ferritin (Section 2), including its effects on AML-supporting bone marrow stromal cells and especially endothelial and immunocompetent cells (Section 3).We then present a discussion of ferritin as one out of several acute-phase biomarkers (Section 4) together with the biological and the possible clinical role of the acute-phase reaction and acute-phase mediators for the initial pretreatment prognostication (Section 5), the regulation of inflammation/coagulation in AML (Section 6) and the prognostication for AML patients receiving allogeneic stem cell transplantation (Section 7).We thereafter review the importance of the iron metabolism for the proliferation/survival/chemosensitivity of AML cells. We then focus on ferroptosis (Section 8) and possible strategies for the therapeutic targeting of the iron metabolism and the regulation of ferroptosis in AML (Section 9).

The last part of our article includes a general and summarizing discussion (Section 10) and a brief concluding comment (Section 11) with regard to the role of ferritin in human AML.

## 2. The Molecular Structure and the Molecular Interactions of Ferritin

Several excellent reviews have described and discussed the structure and general functions of the ferritin molecule [6,7,8,9,10]. We give a brief summary of its most important molecular and biological characteristics in the following sections and in Table 1.

### 2.1. The Molecular Structure, Cellular Expression, Function and Secretion of Ferritin

The structure, expression and functions of ferritin are summarized in Table 1. The ferritin light (L-ferritin, *FTL*) and heavy chains (H-ferritin, *FTH*) will spontaneously form a stable globular structure of 24 subunits [6]. Tetramers, hexamers and dodecamers have also been detected, but these forms are regarded as intermediates [11]. The process of self-assembly possibly starts with the formation of dimers, but monomers and even odd-number intermediates may also participate in the process [6]. The cellular ferritin level is regulated both by transcriptional and post-transcriptional mechanisms, including degradation:

The transcription of especially *FTH* but also *FTL* is increased by proinflammatory cytokines, e.g., Interleukin 1β (IL1β), IL6 and Tumor necrosis factor α (TNFα) through increased NFκB (the nuclear factor kappa-light-chain enhancer of activated B cells) activity; this mechanism is operative in hepatic cells [12,13,14,15].Interferon (IFN) γ and lipopolysaccharide/TLR4 (Toll-like receptor 4) ligation in macrophages increase especially FTH, but also FTL expression through an alternative nitric-oxide-dependent mechanism that involves the degradation of IRP2 (iron-responsive element-binding protein 2) when the cellular iron level is adequate [16,17,18,19]. IRP2 regulates the cellular iron levels by binding to iron-responsive RNA elements and the iron level is thereby increased by modulated translation as well as the stability of mRNAs involved in iron homeostasis.Post-transcriptional regulation is mediated through the iron-responsive elements of ferritin-encoding RNAs; the final effect is then an adjustment especially of FTL, but also FTH levels to the intracellular iron level [18].The cellular ferritin level is also regulated by ferritin degradation through ferritinophagy, i.e., autophagic/lysosomal degradation that is associated with the induction of ferroptosis (see Section 7) [20,21].

A main function of ferritin is to oxidize and store iron. Additional functions include an antioxidant effect by reducing toxic free iron and thereby protecting cells from oxidative stress [5,6]. Ferritin is also a regulator of intracellular iron metabolism as will be described more in detail later (Section 8). Murine studies suggest that there is a 2:1 distribution of intracellular ferritin between membranous compartments and cytosol [22].

Mammalian ferritin lacks the signaling sequence for classical endocytoplasmatic reticulum-Golgi secretion and is, therefore, secreted via alternative pathways, including the multivesicular body–exosome pathway and autophagosome-related pathways [22,23,24]. Ferritin is secreted by various cells (including monocytes and hepatocytes) and can, therefore, be detected in the serum [22,23,24]. The serum form comprises mainly L ferritin subunits, it is poor in iron, and it has been proposed to be derived mainly from macrophages [23,25]. Ferritin can bind to other serum proteins including high-molecular-weight kininogen, apolipoprotein B, α-2-macroglobulin and fibrinogen [18,26,27,28]. The possible functional importance of such binding to other serum/plasma molecules is not known, but the removal of ferritin from the circulation through the receptor-mediated endocytosis of α-2-macroglobulin may be a possibility.

### 2.2. The Cellular Ferritin Receptors for Extracellular Ferritin and Their Expression by AML Cells

Ferritin can bind to various cell surface receptors, as will be discussed below. The functional importance of the various receptors in different cell types, including primary human AML cells, is largely unknown.

*Transferrin receptor 1 (CD71/TFRC/TFR).* This receptor can bind H-ferritin and thereby mediate the cellular uptake of ferritin through endocytosis; ferritin can thereafter be transported to lysosomes for degradation [29,30]. This CD71 transferrin receptor seems important for proliferation and the survival of primary human AML cells [2,3].

*Scavenger receptors.* The possible ferritin interactions with the five members of the Scavenger receptor class A (SCARA) have been investigated [31]. These receptors are transmembrane proteins that form homotrimers on the cell surface, and both SCARA1/CD204, SCARA2 and SCARA5 can ligate ferritin. The ligation of SCARA5 has been characterized more in detail; SCARA5 functions as a ferritin receptor through Ca^2+^-dependent binding of both L- and H-ferritin to a specific site, that seems to be expressed also by SCARA1 and SCARA2. The ferritin ligation of SCARA5 functions as an alternative system for non-transferrin iron delivery, and the SCARA5 ligation then stimulates the endocytosis of this receptor/ligand complex [32].

The effect of SCARA ligation on downstream intracellular signaling has been characterized only for SCARA5. Studies in various normal and malignant cells suggest that the ligation of this receptor modulates the activation of various intracellular pathways [33,34,35,36,37,38], including PI3K (Phosphatidylinositol-4,5-bisphosphate 3-kinase)-Akt(Protein kinase B)-mTOR (the mechanistic target of rapamycin kinase) [34,37,38], FAK (focal adhesion kinase) [35,36], STAT3 (a signal transducer and activator of transcription 3) [34,38] and ERK (extracellular signal-regulated kinase)1/2 activation [38]. However, the final effects of receptor ligation may (at least partly) depend on the cell type, and for the FAK target, both the inhibition and stimulation of its downstream signaling have been described in osteosarcoma and hepatocellular carcinoma cells, respectively [35,36]. Thus, SCARA5 ligation does not only result in iron delivery, but also the modulation of intracellular signaling.

SCARA1/2 as well as SCARA5 can be expressed by normal myeloid cells, especially macrophages, but also by bone marrow stromal cell [33,39]. SCARA5 expressed in fibroblast-derived exosomes can then (indirectly?) inhibit PI3K/AKT signaling by neighboring malignant cells and thereby counteract growth-enhancing effects [39]. Furthermore, SCARA5 is also expressed by endothelial cells, another AML-supporting bone marrow cell type [40]. Finally, SCARA5 ligation by ferritin seems to induce ferroptosis in certain malignant cells [41], but it is not known whether SCARA5 is expressed by or has a role in the regulation of intracellular signaling/ferroptosis in human AML cells (see Section 7).

*Macrophage scavenger receptor 1 (MSR1/CD204).* This receptor can be expressed by myeloid cells. The exposure of neutrophils to ferritin increases MSR1 expression (including its cell surface expression), but ferritin also acts as a MSR1 ligand to trigger extracellular trap formation and cytokine release [10]. The receptor has several other endogenous ligands, including collagen, heat shock proteins, various lipoproteins and certain molecules expressed/released by apoptotic cells [42]. Its initiation of downstream signaling seems to depend on internalization by phagocytosis/endocytosis, and at least certain downstream events partly depend on polyubiquitination [42]. The receptor can thereafter serve as a scaffold to recruit interacting proteins (for additional details and original references, see [42]). Receptor ligation then seems to influence several intracellular pathways including the two pathways SRC (SRC proto-oncogene, non-receptor tyrosine kinase)/Rac1 (Race family small ATPase 1)/Pak (P21 activated kinases)/Junk (c-Jun N-terminal Kinase) and SRC (SRC proto-oncogene, non-receptor tyrosine kinase)/Rac1/Pak/p38; additional interactions with Toll-like receptors lead to the activation of NFκB and IRF (interferon regulatory factor). However, it should be emphasized that most previous studies have been performed only in monocytes/macrophages, and the final functional effect of receptor ligation then seems to depend both on the ligand and the biological context (i.e., the macrophage activation status). These observations suggest that the effect of ferritin on primary AML cells will vary between patients (e.g., degree of AML cell differentiation) and possibly also between cell subsets within the hierarchically organized AML cell population.

MSR1/CD204 can be expressed by primary AML cells [43] as well as cancer- and AML-associated macrophages of the M2 phenotype [33,42,44,45]. To the best of our knowledge it is not known whether this receptor has a role in the induction of AML cell ferroptosis, but studies of various ligands have shown that the receptor/ligand complex is endocytosed following ligation [46]. This last observation suggests a possible role for this receptor in the cellular uptake of ferritin.

*T cell immunoglobulin and mucin domain-containing protein 2 (TIM-2).* TIM-1 is a murine endocytosing receptor for H-ferritin; this has been demonstrated in various cells including oligodendrocytes as well as certain immunocompetent cells [47,48]. The expression of TIM-1 is regulated by iron; iron loading decreases whereas iron chelation increases its expression. The TIM-1 receptor is not expressed in human cells, but the TIM-2 receptor shows structural similarities with TIM-1 and is expressed by human cells [48].

*C-X-C chemokine receptor type 4 (CXCR4).* This receptor seems able to bind and internalize the heavy ferritin chain [49]. The binding of ferritin-H inhibits its ligand-induced downstream activation of ERK1/2 [37,49]. This receptor is important for the crosstalk between AML cells and their neighboring leukemia-supporting stromal cells in the bone marrow microenvironment [50] and the targeting of this receptor or its downstream signaling is regarded as a possible therapeutic strategy in AML [51].

## 3. Microenvironmental Effects of Extracellular Ferritin Release: Possible Effects on AML-Associated Angiogenesis and Immunoregulation in the Bone Marrow

### 3.1. The AML Cells and Their Bone Marrow Microenvironment

AML is a bone marrow disease (Figure 1) where the leukemic cells communicate with neighboring nonleukemic and AML-supporting cells through direct cell–cell contact and the extracellular release of soluble mediators. Several nonleukemic/nonhematopoietic cells support normal as well as leukemic hematopoiesis, including osteoblast and endothelial cells that form specialized vascular and endosteal stem cell niches, respectively [52]. Other bone marrow cells can also support leukemogenesis, including osteolineage cells, mesenchymal stem/progenitor cells, adipocytes, perivascular cells, megakaryocytes, various immunocompetent cells (monocytes, neutrophils and T cells) and the neural innervation [52].

The functional effects of soluble ferritin on the majority of these AML-supporting cells are largely unknown. The bone marrow infiltration of various immunocompetent cells can support leukemic hematopoiesis, e.g., through their release of hematopoietic growth factors. A previous study based on the gene expression profiles of AML bone marrow samples suggested that the infiltrating immunocompetent cells are very heterogeneous and include CD4^+^ T cells, neutrophils, macrophages monocytes, dendritic cells, natural killer (NK) cells, myeloid-derived suppressor cells (MDSC), regulatory T cells (Treg) and immature B cells [53]. Another study based on the same methodological strategy [54] suggested that patients with high-risk AML had a higher level of infiltration of immunocompetent cells, especially AML-supporting M2 macrophages, γδ T cells as well as immunosuppressive MDSCs and Treg cells. Finally, angiogenesis is regarded as a fundamental process in leukemogenesis and for chemosensitivity in AML (see Section 3.3), and soluble ferritin seems to influence endothelial cells and possibly also their dual AML-supporting function mediated by (i) increased bone marrow angiogenesis and (ii) increased AML-supporting effects mediated by vascular/endothelial stem cell niches.

### 3.2. Ferritin and AML-Supporting Nonleukemic Cells in the Bone Marrow Microenvironment: Ferritin Effects on Immunocompetent Cells

Ferritin has immunomodulatory effects, but it should be emphasized that many of these effects have been detected only in experimental studies and/or animal models (for additional references, see [29,55]):Ferritin can inhibit normal hematopoiesis, including myelopoiesis (i.e., granulocyte/monocyte progenitors) [56,57,58,59].Ferritin seems to inhibit mitogenic T cell activation [60].The binding of ferritin H to the CXCR4 chemokine receptor can inhibit ligand-induced downstream ERK1/2 activation [49].Delayed-type hypersensitivity can also be inhibited by ferritin in animal models [61]; this effect is possibly mediated by the induction of IL10 [62].Ferritin seems to alter the profile/distribution of circulating T cells and to suppress T helper cell activity [63].Ferritin may also inhibit the function of tumoricidal monocytes/macrophages [64].Ferritin seems to have immunosuppressive effects through the modulation of dendritic cell functions and thereby the activation of Treg cells [65].Experimental animal studies suggest that even distant disease (i.e., spinal cord injury) can alter iron metabolism in normal cells, including the ferritin function and iron metabolism in macrophages [66].

Ferritin-L can be upregulated and promote the M2-polarization of cancer-associated macrophages and this M2-polarization can both facilitate local angiogenesis and reduce local T cell recruitment [67]. Furthermore, AML-supporting M2 macrophage subsets are increased both in preclinical models of AML and in AML patients [68]. Cytokines derived from T cells [69] (including IL4/IL10/IL13) and AML cells [70] can possibly contribute to M2 macrophage differentiation [68,70] and facilitate a switch from M1- to M2-like macrophages that support AML cells through various mechanisms [71,72,73], including a mitochondrial transfer leading to metabolic reprogramming [73]. Bone marrow T cells may thus support leukemogenesis indirectly by supporting the development of M2-like macrophages and directly through the release of growth factors for the AML cells (e.g., GM-CSF, IL3) [69].

Several studies of gene expression profiles of AML bone marrow have investigated the prognostic impact of M2 macrophage infiltration. A high expression of M2-associated CD206 was then associated with inferior overall as well as event-free survival in AML [71], but the use of this marker alone is not reliable because CD206 can be expressed also by immunosuppressive dendritic cells [74] and even AML cells [75]. However, another study showed that decreased survival was also associated with the alternative M2 marker CD163 [72,74]. Thus, the high infiltration of M2-like bone marrow macrophages are associated with clinical chemosensitivity and adverse prognosis in human AML [72,76].

### 3.3. Ferritin and AML-Supporting Nonleukemic Cells in the Bone Marrow Microenvironment: Ferritin Effects on Endothelial Cells and Bone Marrow Angiogenesis

Several studies suggest that ferritin can modulate the functional characteristics of endothelial cells, and thereby it may also become involved in the regulation of angiogenesis:Ferritin is regarded as a cytoprotective molecule that has an antioxidant function and sequesters free cytoplasmic iron in endothelial cells [77,78]. Endothelial ferritin levels can then be increased in vitro by iron, exogenous ferritin and aspirin [77,78].Mitochondrial ferritin can also be upregulated by cellular stress; it then has an antiapoptotic effect with the prevention of tight junction loss by reducing iron dysregulation and the accumulation of reactive oxygen species (ROS) [79].Toxic effects on endothelial cells can include ferritinophagy followed by ferroptosis [80]; these observations further support the hypothesis that ferritin is important for the endothelial cell response to stress, possibly including antiangiogenic therapy.High-molecular-weight kininogen is a coagulation cofactor that can be cleaved by serine proteases into the proangiogenic bradykinin (nine amino acids) and an antiangiogenic cleavage product [81,82,83]. Ferritin can bind to high-molecular-weight kininogen and thereby inhibit this protease cleavage [84,85]. However, the ferritin binding site is on the antiangiogenic cleavage product, and ferritin may, therefore. counteract the proangiogenic but not the antiangiogenic effect of the cleavage [86].

Taken together these observations suggest that ferritin can modulate endothelial cell functions/survival and thereby act as a regulator of local angiogenesis and vascular remodeling, possibly also AML-induced angiogenesis. These preliminary observations suggest that possible effects of ferritin on the leukocyte chemotaxis/transendothelial migration of antileukemic immune reactivity [87,88,89] and AML-supporting macrophages [53,54] as well as endothelial cell support including local AML-induced angiogenesis [90,91,92] should be further investigated.

### 3.4. Effects of Ferritin on Adipocytes: Are the Effects Relevant in AML?

Several studies have shown that ferritin can modulate differentiation, energy metabolism and the epigenetic regulation of adipocytes [93,94,95]. Bone marrow adipocytes function as regulators of normal hematopoiesis; these cells may also regulate leukemic hematopoiesis in AML [96], but this possible role needs to be further investigated.

### 3.5. Effects of Iron and Iron Overload on Mesenchymal Stem Cells

The bidirectional crosstalk between AML cells and mesenchymal stem cells (MSC) supports AML cell proliferation/survival and modulates the functions of MSCs [97,98,99]. Several studies have also demonstrated that AML-relevant MSC functions are altered by iron and/or iron overload, as summarized in Table 2 [100,101,102,103,104,105,106,107]:The release of several cytokines is reduced, and these modulations can have both direct effects on the AML cells and indirect effects via immunocompetent or endothelial cells (angiogenesis and vascular stem cell niches).Reduced osteoblastic differentiation can influence the support of endosteal stem cell niches.AML cell proliferation/survival are supported both by endothelial [108] and osteoblastic cells [109]. This is similar to normal hematopoiesis [52], but additional studies are needed to clarify whether the AML-supporting effects are altered for iron-modulated MSCs.The ability to support normal hematopoiesis is reduced and this observation supports the hypothesis that leukemic hematopoiesis may also be affected.

To conclude, iron or iron overload modulate several AML-relevant MSC functions in experimental studies, but the final overall effects on AML leukemogenesis and their possible clinical relevance are difficult to predict.

## 4. The Acute-Phase Reaction in AML: A General Description of Acute-Phase Biomarkers (Including Ferritin) That Have a Clinical Relevance in Human AML

### 4.1. General Aspects of the Acute-Phase Reaction: Altered Systemic Levels of Several Molecular Markers Including Ferritin as Well as Altered Levels of Circulating Normal Blood Cells

The term acute-phase reaction is used to describe inflammation-induced effects distant from the site of inflammation; these effects include altered levels of a large number of plasma proteins as well as other biochemical markers together with cellular and nutritional changes [110]. It should be emphasized that such alterations can be observed not only in acute, but also during chronic inflammation.

An acute-phase protein has been defined as a protein showing at least a 25% increase or decrease in the systemic plasma/serum concentration during inflammation [110]. The changes vary considerably between acute-phase proteins; CRP (C-reactive protein) and ferritin can show usually 10–100-fold but even up to 1000-fold increases, fibrinogen usually shows 2–10-fold increases and other proteins only up to a 2-fold increase [110,111,112]. Many other proteins can also be classified as acute-phase proteins according to their variation, for example, in AML patients with febrile neutropenia, including several interleukins and chemokines, hematopoietic growth factors (e.g., G-CSF, GM-CSF and thrombopoietin), various proinflammatory cytokines (IL1β, IL6, IL8/CXCL8, TNFα and IFNγ) as well as soluble cytokine antagonists/receptors (i.e., IL1RA, soluble TNF receptors and soluble IL4Rα), soluble adhesion molecules (e.g., selectins, ICAM-1 and CD14) and soluble matrix molecules (e.g., endocan) [113]. Various proinflammatory cytokines contribute to the initiation of an acute-phase reaction; there is usually an interaction between various cytokines, and the dominating initiating mechanism can vary depending on the cause of the inflammation [110,113]. An overview of various acute-phase biomarkers used in routine clinical practice are given in Figure 2.

Many patients with severe diseases have nutritional defects and AML patients can show signs of malnutrition both before and during treatment [114,115]. The nutritional status will also influence the acute-phase reaction and possibly weaken this response [116]. Finally, elderly individuals may have signs of an acute-phase reaction and/or anemia [52]; this is referred to as inflammaging and a disease-associated reaction (e.g., in elderly AML patients) will then develop in addition to this predisease reaction.

### 4.2. Serum Ferritin Levels: Hematological Malignancy Is an Uncommon Cause of Hyperferritinemia

Ferritin is expressed in most tissues as a cytosolic protein and is released in small amounts into the serum; its main function is the storage of iron in a soluble and nontoxic form [117]. Several tissues may thus contribute to increased systemic ferritin levels.

Hyperferritinemia is often defined as the total serum ferritin exceeding 200 μg/L in women and 300 μg/L in men; there is no general agreement for the definition of extreme hyperferritinemia and previous studies have used values ranging from 2000 to 10,000 μg/L [118,119,120]. A previous study from three French university hospitals included adult patients with high ferritin levels, defined as serum ferritin exceeding 5000 μg/L and extreme hyperferritinemia was then defined as exceeding 25,000 μg/L [118]. They identified 495 patients with hyperferritinemia; their mean age was 57 years, the median ferritin level was 9128 μg/L (range 5013–1,133,280 μg/L), median CRP was 104 mg/L (range 1–614 mg/L) and 78% of the patients had a history of cancer. Furthermore, 388 patients were also classified as immunocompromised, i.e., being transplanted, having a malignancy (mainly hematological), receiving chemotherapy during the last six months, having an HIV infection or receiving long-term immunosuppressive therapy. The causes of hyperferritinemia are listed and classified in Table 3. It should be emphasized that 81% of their patients had more than one possible cause.

It can be seen from the results presented in Table 3 that a wide range of various diseases can have serum ferritin levels exceeding 5000 μg/L, including infections, malignancies, rheumatic/inflammatory diseases, liver diseases and hemolysis, whereas iron overload constitutes a relatively small minority of the patients. However, extreme hyperferritinemia exceeding 25,000 μg/L was mainly caused by hemophagocytic lymphohistiocytosis (HLH), with other causes being cytokine release syndrome, acute hepatitis or infections. Serum ferritin levels exceeding 250,000 μg/L were only seen for patients with HLH (nearly 90% of cases) or cytokine release syndrome. However, it should be emphasized that the ferritin levels alone have a limited impact with regard to the diagnosis of HLH [119,120,121]. The observations from this study seem to be relevant for patients with hyperferritinemia in general [119,120].

Some studies of ferritin levels use CRP correction of the levels, i.e., ferritin levels are corrected by using a correction factor of 0.67 if there is a concomitant increase in the CRP levels [122]. However, most studies of hyperferritinemia do not use this correction, which is mainly used to evaluate patients with a suspected iron deficiency.

Ferritin/iron metabolism is important both for immunomodulation (see Section 3.2) [67] and the regulation of cell survival [123,124]. Both these effects may be relevant in AML patients with increased ferritin. First, ferritin can reprogram the immune microenvironment of tumors, and similar mechanisms may also be operative in the AML-supporting bone marrow microenvironment where various immunocompetent cells (including macrophage subsets) seem to support the leukemic cells [53,54,68,71,72,73,125]. Second, ferroptosis is a form of programmed cell death that seems important for chemosensitivity in human AML cells [126,127].

### 4.3. CRP in the Acute-Phase Reaction: Increased Levels Not Only During Infections, but Also in Inflammation, Malignancies and Aging

The structure and molecular function of CRP has been described in several recent reviews (for references, see [113]). Briefly, CRP exists in various isoforms, the basic molecular unit being the monomer that consists of 206 amino acids. The main serum form is a pentamer derived from hepatocytes. The effects of this pentamer on monocytes/macrophages are mediated by various complement receptors, whereas effects on endothelial cells are mediated by lipoprotein receptors and effects on platelets by α_v_β_3_ integrins or other CRP-binding integrins.

The infusion of recombinant human CRP results in increased levels of proinflammatory cytokines, an observation suggesting that CRP is not only a biomarker, but also a mediator in inflammation [113,128]. CRP can alter the function of several immunocompetent cells (for additional details and references, see [113]). First, it can stimulate the release of proinflammatory soluble mediators by monocytes and support polarization towards the proinflammatory M1 phenotype. Second, CRP can activate dendritic cells and thereby stimulate T cell activation. Third, it can activate endothelial cells. Finally, other immunomodulatory CRP effects include complement activation, the inhibition of neutrophil activation/chemotaxis and increased platelet activation. Several of the affected cells may influence both leukemogenesis and susceptibility to conventional intensive chemotherapy, including stem cell transplantation, e.g., through the release of immunomodulatory cytokines by various immunocompetent cells [129,130] (see below).

### 4.4. Decreased Albumin in the Acute-Phase Reaction: Inflammation, Malnutrition and Malignancy Can All Be Associated with Hypoalbuminemia

Albumin is synthesized in the liver before being released to the circulation; decreased serum albumin levels are observed in inflammation and seems to be due mainly to albumin loss/degradation rather than decreased synthesis [131]. Furthermore, inflammation is often associated with increased vascular permeability with the extravasation of albumin. VEGF seems to be involved in the development of this increased permeability, but the possible contribution of other antiregulatory cytokines has not been investigated in detail. Extravasated interstitial albumin can then serve as a local scavenger protein or as an antioxidant before it returns to the circulation and is degraded in the liver where albumin degradation is increased during inflammation. Alternatively, interstitial albumin is also internalized by local cells and serves as a source of amino acids, e.g., in malnutrition. Thus, increased breakdown in the liver and peripheral tissue seems to be the most important mechanisms for the development of hypoalbuminemia in inflammation.

Primary AML cells usually show the constitutive release of several angioregulatory cytokines that are important in the bidirectional crosstalk between AML cells and endothelial cells in the bone marrow microenvironment [4]. This crosstalk alters the functional characteristics of both cell types and supports AML cell proliferation [108,132,133,134], and the AML-induced modulation of endothelial cells includes the altered expression/release of adhesion molecules [134,135,136]. Studies in xenograft models have also demonstrated that AML is associated with increased microvascular permeability in the bone marrow [137]. Thus, increased local vascular permeability may then be a local mechanism that contributes to the development of hypoalbuminemia in AML, but it is not known whether AML results in increased permeability also in distant organs.

### 4.5. Fibrinogen Levels in Cancer Patients Can Be Modulated Both by Inflammation and by Coagulopathy

Fibrinogen is a glycoprotein produced by the liver and plays an important role in the regulation of coagulation and hemostasis [138]. It is enzymatically converted by thrombin to fibrin during vascular injury. However, fibrinogen is also an acute-phase protein that can show considerably increased serum levels during inflammation [139,140] and inflammation can in addition shift the overall hemostatic balance toward a prothrombotic state, e.g., during severe COVID-19 infections [141]. Furthermore, systemic levels of proinflammatory cytokines (e.g., IL-6, TNF-α) are increased during the acute-phase reaction and stimulate the liver to produce and release fibrinogen [142]. Fibrinogen can then function as a regulator of inflammation through effects on various immunocompetent/inflammatory cells, including the stimulation of leukocyte transendothelial migration as well as the enhancement of leukocyte phagocytosis and cytokine release [143]. Thus, fibrinogen serves as a biomarker, an immunoregulator and a procoagulant in inflammation.

Fibrinogen consists of two sets of three polypeptides termed Aα, Bβ and γ that are joined together by disulfide bonds [144]. Different forms of the molecule originate from alternative splicing and post-translational modulation that can occur during synthesis or after its secretion [145]. Thus, circulating fibrinogen includes various molecular forms; relative proportions of various forms seem to differ between inflammatory settings, but the total level also increases with age [145]. Fibrinogen is mainly expressed in and secreted by hepatocytes, but small amounts are also detected in non-hepatic tissue [146].

Fibrinogen can bind ferritin (see Section 2.1) as well as several β2 integrins expressed by various myeloid cells including AML cells; fibrinogen thereby becomes important for the binding of these cells to other cells, including their transendothelial migration [125,139]. Fibrinogen may similarly regulate AML cell homing/migration, and it may also influence cellular iron metabolism through its binding to ferritin. Finally, a recent study showed that AML cells (especially AML cells with an undifferentiated phenotype) can express the fibrinogen Aα and Bβ chains as well as the fibrinogen receptor, [147]. However, future studies have to clarify whether fibrinogen has a clinically relevant role in the homing, leukemogenesis or chemoresistance of AML.

### 4.6. Effects of Inflammation on the Systemic Levels of Other Biomarkers of Iron Metabolism: Do These Levels Have Any Clinical Relevance in AML?

As described above, systemic ferritin levels can be increased during inflammation (Section 4.1). However, hepcidin levels are also increased during acute-phase reactions, whereas transferrin levels are decreased [148]. It is not known whether these effects have any clinical relevance in AML, e.g., for the chemosensitivity during antileukemic therapy and/or in the immunoregulation of pancytopenic AML patients receiving intensive chemotherapy or allogeneic stem cell transplantation (Section 5.1 and Section 7.1).

## 5. The Pretreatment Levels of Ferritin and Other Acute-Phase Biomarkers in Patients with Newly Diagnosed AML: Associations with Prognosis After Intensive Therapy

### 5.1. Serum Ferritin Levels in Newly Diagnosed AML Is Associated with Prognosis

Primary AML cells can synthesize ferritin; these cellular levels seem to be higher than the levels in normal leukocytes as well as chronic myeloid leukemia (CML) cells [149,150], but ferritin derived from AML cells seems to have low iron binding [149]. The pretherapy serum ferritin levels seem to be higher in AML than CML patients [150]. Remission induction can then reduce the ferritin levels [151,152]. Thus, AML cell release can contribute to the increased ferritin levels in patients with untreated AML, and this contribution will then decrease after remission induction.

A recent systematic review and meta-analysis evaluated previous clinical studies of pretherapy serum ferritin levels for AML patients at the first time of diagnosis [153]. The authors identified nine studies that fulfilled their criteria. Their analyses revealed a statistically significant association between high pretreatment serum ferritin levels (i.e., serum ferritin > 1000 µg/L) and poor outcomes with lower overall survival and event-free survival as well as an inverse correlation between high ferritin and the onset of infections (i.e., indicating an increased risk of early toxicity/mortality). The observations from the three largest studies identified in this review are described more in detail below.

The first retrospective single-institution study examined pretreatment/baseline serum ferritin levels in 137 patients (median age 54 years, range 43–65 years) receiving induction treatment based on anthracycline plus cytarabine [154] This group represents all patients with an available baseline ferritin level during a defined time period; 85 patients had de novo AML and 46 had an adverse karyotype. Seventy-six patients received consolidation with high-dose cytarabine, whereas 46 patients received allogeneic stem cell transplantation. After the CRP-based adjustment of the ferritin levels (see Section 4.2, [122]), patients were grouped according to their pretreatment level. High ferritin levels, defined as serum ferritin ≥ 750 μg/L (corresponding to an uncorrected level of 1120 μg/L), were associated with significantly lower overall and relapse-free survival and high baseline levels remained an independent adverse prognostic factor also in the multivariate analysis. High ferritin was associated both with a higher relapse risk and higher non-relapse mortality. These authors also published a later and larger retrospective clinical study of 590 elderly AML patients. For these patients, they observed that the association between high serum ferritin levels and adverse prognosis was strongest for patients receiving intensive treatment, whereas strong associations were not observed for patients receiving AML-stabilizing therapy or the best supportive care [155].

A second multicenter retrospective study evaluated the clinical significance of pretreatment serum ferritin in 305 newly diagnosed AML patients receiving induction therapy based on anthracycline plus cytarabine [156]. The patients had a median age of 50 years (range 15–77 years), half of the patients had an intermediate karyotype, 168 patients received allogeneic stem cell transplantation and the median follow-up time was 58 months (range, 4–187 months) for the survivors. The median ferritin value was 512 μg/L (range, 8–9475 μg/L) and ferritin correlated with lactate dehydrogenase, CRP and the white blood cell/blast count. Increased ferritin was also associated with a poor performance status. Patients with ferritin ≥ 400 μg/L showed inferior 5-year event-free survival compared with the low ferritin group (30% versus 40%; *p* = 0.033). The multivariate analysis of patients with the high-risk karyotype (96 patients) showed that high ferritin levels predicted worse event-free survival even for this group (hazard ratio = 2.07; *p* = 0.003).

The third study included 162 AML patients (median age 49 years, range 16–60 years) with the intermediate-risk karyotype (111 with the normal karyotype) receiving intensive chemotherapy [157]; 146 patients achieved complete hematological remission. The median ferritin level at diagnosis was 633 μg/L and 128 patients (79%) had pretreatment ferritin levels above the upper normal limit. High pretreatment ferritin levels (i.e., higher than four times the upper normal limit) were significantly associated with a higher relapse incidence and decreased AML-free as well as overall survival. High ferritin levels remained an independent adverse prognostic factor in multivariate analysis. A weaker association was also seen between CRP > 55 mg/L and AML-free survival (*p* < 0.0001 for ferritin versus *p* = 0.006 for CRP), whereas CRP showed no statistically significant association with overall survival.

Taken together, these studies suggest that increased/high serum ferritin levels at the time of first diagnosis have a prognostic impact for AML patients receiving potentially curative therapy based on intensive anthracycline + cytarabine induction chemotherapy. The high ferritin levels at diagnosis cannot be explained by previous transfusions and this seems to be true even for patients with secondary AML [154]. The difference between CRP and ferritin with regard to the prognostic impact [157] and the remaining prognostic impact after a CRP-adjustment of ferritin levels [154] suggests that the presence of AML-associated subclinical inflammation alone cannot be the only explanation for the prognostic impact of ferritin.

Possible mechanisms behind the association between ferritin and prognosis could be (i) ferritin-induced immunomodulation with altered AML-supporting effects by certain immunocompetent cells in the bone marrow microenvironment (see Section 3.2), (ii) altered AML-supportive effects by endothelial cells either through increased local angiogenesis or the modulation of vascular stem cell niches (see Section 3.3) or (iii) ferritin-induced alterations in iron metabolism/ferroptosis in the AML cells (see Section 2.2). Furthermore, increased long-term AML-free survival is associated with both high general constitutive cytokine release as well as altered cellular iron metabolism of the AML cells [4]; these two observations together further support the hypothesis that both AML cell (possibly cytokine-mediated) communication with neighboring non-leukemic supporting cells and the AML cell iron metabolism (possibly modulated by ferritin) are important for this ferritin-associated clinical chemosensitivity of primary AML cells. Finally, altered iron metabolism and/or increased ferritin levels seem to be a part of a high-risk phenotype also in other malignancies; the observations in AML may thus reflect a ferritin effect that is not AML-specific, but also operative in other malignancies [158,159,160].

### 5.2. Serum CRP, Albumin and Fibrinogen Levels in Patients with Newly Diagnosed AML

Several retrospective studies have investigated pretreatment serum/plasma levels of other acute-phase biomarkers than ferritin at the first time of AML diagnosis. The associations between these alternative acute-phase biomarkers and overall survival, event-free survival and nonrelapse mortality are summarized in Table 4 [161,162,163,164,165,166,167,168,169]. This presentation is based on selected studies including relatively large numbers of patients (i.e., the smallest study including 188 patients) and the studies examined the well-established acute-phase biomarkers CRP, albumin and/or fibrinogen. Even though different acute-phase biomarkers and different cut-off levels were used in the various studies, they all described significant associations between high levels of the investigated biomarkers and at least one of the three clinical events/endpoints. The most common significant association was between the acute-phase parameters and long-term overall survival; this association was observed for all eight studies investigating long-term survival and thus both for elderly and younger patients. Significant associations with AML- or event-free survival were also observed for all except three studies; two of the exceptional studies included many/mainly elderly patients [163,165]. A significant association with nonrelapse mortality was observed only for a large albumin study including 756 patients with a high median age of 60 years [163], but two studies observed increased early toxicity/mortality for patients with increased acute-phase biomarkers [161,163]. Thus, signs of an acute-phase reaction (similar to high ferritin levels) seems to be associated with increased mortality due to an increased risk of relapse and probably also an increased risk of severe/lethal toxicity.

### 5.3. Possible Mechanisms Behind the Association Between High Pretreatment Ferritin Levels, Increased Acute-Phase Biomarkers and Adverse Prognosis After Intensive AML Chemotherapy

We have summarized the associations between increased pretreatment ferritin levels and selected clinicobiological parameters as well as the levels of other acute-phase markers (Table 5) that are also associated with adverse AML prognosis (see also Table 4 above). It should be emphasized that these observations must be interpreted with great care because all studies did not include all comparisons (this is marked as not tested in Table 5).

The following conclusions/suggestions can be made based on the observations that are summarized in Table 5 (see above):

These relatively large clinical studies are heterogeneous with regard to investigated correlations between the levels of ferritin and other acute-phase marker(s), other adverse prognostic factors and patients’ ages (one study investigating only elderly patients [165]). However, despite these heterogeneities, they all show adverse prognosis for patients with high levels of various acute-phase markers (Table 5).

*Age:* Ferritin did not show any significant association with the patients’ ages in any study; this was true also for fibrinogen and FAC ratio studies, whereas an association with age was seen in CRP–albumin studies. The association between age and acute-phase biomarkers other than ferritin may be caused by the previous presence of inflammaging that can be observed in elderly patients, i.e., signs of inflammation often associated with reduced physical functions [52]. Inflammation may thereby contribute to the association between CRP–albumin markers and patient age at the time of first AML diagnosis, but if this is true, inflammaging seems less important for the increased ferritin level in AML.*Sex:* All three ferritin studies showed a significantly higher frequency of males among patients with high ferritin [154,156,157]; an opposite observation was made in one study of the CRP–albumin ratio [165], but for the other studies, the ratio did not differ [161,162,163,164,166]. Further studies are needed to explain these differences.*Associations between ferritin and other acute-phase biomarkers:* Despite differences between various acute-phase biomarkers with regard to associations with age and sex, several studies described significant correlations between the systemic ferritin levels and the other acute-phase biomarkers. These observations are consistent with the hypothesis that the prognostic impact of ferritin in AML is partly caused by its association with inflammation and an acute-phase reaction.*Secondary AML:* Another difference between various acute-phase markers was their association with secondary AML; such an association was only observed for some CRP–albumin studies, but not for other markers. The association between secondary AML and CRP–albumin may be caused by inflammation/inflammatory complications that can be seen in patients with myelodysplastic syndromes (MDS) [170,171,172,173,174]. Furthermore, the absence of any association between the ferritin level and secondary AML suggest that the contribution of an iron overload due to previous transfusions, as would be expected for patients with previous MDS or cancer therapy, is less important for the increased ferritin levels at the time of first AML diagnosis. An association between high ferritin levels and secondary AML was not associated even in the study by Ihlow et al. [154] that included a relatively large number of patients with secondary AML. Thus, MDS-associated inflammation and previous transfusions are possibly less important for the prognostic impact of ferritin in AML.*Leukemia burden:* the associations between leukemization/marrow blast levels suggest that the AML burden contributes to the prognostic impact of various acute-phase biomarkers.*Differentiation:* AML cell differentiation/FAB classification does not show strong associations with acute-phase biomarkers.*Genetic abnormalities:* One study investigated *FLT3* and *NPM1* mutations and described a decreased frequency of *NPM1* mutations in patients with high ferritin [157]. A relatively high frequency of *Flt3-ITD* was also observed in patients with a high CFA ratio [168]. The associations between cytogenetic/molecular genetic abnormalities may reflect differences with regard to the AML induction of inflammation [161,164,167,169].*Performance status:* One albumin study investigated the performance status and described a significant inverse correlation between a high ECOG (Eastern Cooperative Oncology Group) score and albumin level. This observation may also at least partly reflect an effect of inflammaging on ferritin levels [52,175].

To conclude, the acute-phase reaction contributes to increased pretreatment ferritin levels in AML. However, other mechanisms are possibly also involved and this can explain why the prognostic impact of ferritin is independent of or reaches higher significance than CRP in certain studies [157]. The AML cell phenotype, its ability to induce inflammation and the nature of the inflammatory reaction may also be important for the prognostic association (see Section 6.1).

### 5.4. Systemic Levels of Hepcidin in AML

Hepcidin is a 25-amino acid peptide that is produced mainly in the liver. It downregulates cellular iron export by binding to the ferroptin iron exporter and thereby inducing its internalization [176,177]. Serum hepcidin levels can be increased in AML patients at the first time of diagnosis [178,179]. One study described a decrease in these levels after remission induction [179]; these authors also described that the serum levels of soluble transferrin receptors decreased after remission induction and they described a negative correlation between the post-therapy hepcidin levels and markers of erythropoiesis (hemoglobin, reticulocyte and soluble transferrin receptor levels). Furthermore, pretransplant hepcidin levels can still be increased for patients in remission, but normalization can then be seen within one month after allogeneic stem cell transplantation [180]. High pretransplant levels seem to be associated with an increased risk of bacterial and fungal infections post-transplant [180]. Thus, both the AML disease and various antileukemic therapies seem to have complex effects on iron metabolism, but it is not known whether or how pre- or post-transplant serum hepcidin levels will interact/influence the prognostic association/impact of serum ferritin neither at the first time of diagnosis nor in allotransplant recipients (see Section 7).

## 6. Inflammation in Patients with Recently Diagnosed AML: Acute-Phase Reaction/Inflammation, Hemophagocytosis and Coagulopathy

### 6.1. Hemophagocytic Lymphohistiocytosis in AML

The diagnosis of hemophagocytic lymphohistiocytosis (HLH) is usually based on fulfilling at least five out of the eight criteria: (i) fever, (ii) splenomegaly and (iii) peripheral blood cytopenia affecting at least two lineages (hemoglobin < 10.0 g/100 mL, platelets < 100 × 10^9^/L and neutrophils < 1.0 × 10^9^/L), (iv) hypertriglyceridemia or hypofibrinogenemia, (v) hemophagocytosis in bone marrow, liver or lymph nodes, (vi) hyperferritinemia > 500 μg/L, (vii) impaired NK (natural killer) cell function and (viii) increased soluble IL2 receptor [181,182,183]. However, these criteria are difficult to use in AML because these patients are prone to bacterial and fungal infections [184,185,186], have reduced levels of circulating normal cells due to the disease or the chemotherapy [184,185,186,187], the disease itself may cause splenomegaly [188,189], fibrinogen may be increased due to an acute-phase reaction [190] (see Section 5.2), they often receive nutritional support that makes it difficult to interpret the triglyceride level [191], hemophagocytosis may be seen without HLH [192] and increased ferritin levels may have other causes including an iron overload.

The more recently developed Hscore uses partly overlapping criteria and is based on (i) the ferritin level together with (ii) the presence of immunosuppression, (iii) fever, (iv) hepatosplenomegaly, (v) peripheral blood cytopenias, (vi) an increased triglyceride level, (vii) increased fibrinogen level, (viii) increased ASAT level and (ix) hemophagocytosis in the bone marrow [193,194,195,196]. However, the HScore is also difficult to use in AML patients for the same reasons as described for the classical HLH-2004 criteria.

A recent study investigated bone marrow samples derived from 243 AML patients (age ranging from 41 to 69 years) receiving induction therapy; most of them had de novo AML and received treatment based on cytarabine plus anthracycline [192]. The bone marrow samples were derived (i) at diagnosis, (ii) 15 days after the start of induction therapy and (iii) after 35 days if prolonged cytopenia and/or if symptoms/signs consistent with HLH occurred and could not be explained by other causes. They identified 54 patients with hemophagocytosis. Thirty-two of them, in addition, had unexplained fever, increased serum ferritin or unexplained cytopenia and these patients were referred to as hemophagocytosis^+^HLH^+^. The other 22 patients were referred to as hemophagocytosis^+^HLH^−^. The serum ferritin levels were significantly higher for the 32 HLH^+^ patients (median 5095 μg/L) compared with the 22 HLH^−^ patients (median 1867 μg/L, *p* = 0.0004) with only bone marrow hemophagocytosis without additional symptoms. The 32 HLH^+^ patients were also characterized by:Increased serum CRP levels (median 116 versus 19 mg/L, *p* = 0.0005);Increased frequency of hepatomegaly (7/32 versus 0/22, *p* = 0.0335);Respiratory symptoms (19/32 versus 2/22, *p* = 0.0002);Prolonged prothrombin time (median 84 versus 66.5 s, *p* = 0.0013);Decreased albumin levels (median 27 versus 33 g/L, 0.0005);Increased frequency of serum liver transaminases exceeding five times the upper normal limit (7/32 versus 0/22, *p* = 0.0335), increased serum alkaline phosphatase (median 427 versus 214 IU/L, *p* = 0.0005) and increased γGT (median 213 versus 60 IU/mL, *p* = 0.0001).

The observations described above should be interpreted with great care and need verification, but some conclusions can be made. First, an inflammatory response showing similarities with the inflammation in HLH can be detected in approximately 10% of AML patients at diagnosis or during/following the induction treatment. Second, this response partly overlaps with an acute-phase reaction, i.e., showing increased CRP, a decreased albumin level and increased ferritin levels. Third, this inflammatory response is characterized by organ involvement, especially respiratory symptoms, liver involvement and prolonged neutropenia/thrombocytopenia. Finally, bone marrow hemophagocytosis can be seen without the additional signs of HLH/inflammation.

The hemophagocytic characteristics as defined above were associated with decreased survival, especially death in cytopenia [192], and the observations also support the hypothesis that increased ferritin levels at least for some AML patients reflect a specific AML-associated inflammatory phenotype. However, these observations need to be verified in additional larger studies and the possible therapeutic consequences of detecting this Hemophagycytic^+^HLH^+^ syndrome need additional clinical studies.

### 6.2. Pretreatment Coagulopathy in AML: Risk of Thrombosis and the Acute-Phase Reaction

Many AML patients have signs of coagulopathy at the time of diagnosis. The diagnostic criteria for disseminated intravascular coagulation (DIC) is often based on decreased platelet counts, increased fibrin degradation products/D-dimer levels, a prolonged prothrombin time and decreased fibrinogen levels [197]. Based on such criteria, a previous study described that approximately one third of AML patients had coagulopathy consistent with DIC and coagulopathy was also associated with increased CRP levels, i.e., signs of an acute-phase reaction [198]. Coagulopathy may also be associated with monocytic AML cell differentiation [199] and high peripheral blood blast counts [200].

It is not easy to use the four conventional DIC criteria [197] for AML patients because thrombocytopenia due to bone marrow failure is common [2,184,185,186] and almost all patients do not have severe hypofibrinogenemia consistent with DIC [199] possibly because of a concomitant acute-phase reaction that increases fibrinogen levels and thereby counteracts DIC-associated consumption. Furthermore, AML patients have an increased risk of venous and arterial thrombosis, with the risk being estimated to approximately 10% and being highest for elderly patients [199,201]. Most thromboses (66%) occur early before the second chemotherapy cycle [199]; the D-dimer levels alone seem most predictive for thrombosis and there is no significant association between serum CRP levels and risk of thrombosis [199] even though coagulopathy is also associated with an acute-phase reaction/CRP level [198]. A recent review also described several factors that seem to be involved in the pathogenesis of AML-associated thrombosis AML [202], but it is not known whether coagulopathy and/or thrombosis-associated inflammation contribute to the acute-phase reaction and increased pretreatment ferritin levels in AML.

## 7. The Acute-Phase Reaction in Patients Receiving Allogeneic Stem Cell Transplantation

### 7.1. The Prognostic Impact of High Pretransplant Serum Ferritin Levels

Several studies suggest that high pretransplant levels of serum ferritin are associated with adverse prognosis in allotransplant recipients; for additional details/references and a general discussion, we refer to previous reviews and meta-analyses [203,204,205,206]. This prognostic impact seems to be present for patients with AML and preleukemic MDS (Table 6) [207,208,209] as well as other allotransplant recipients (Table 6) [210,211,212,213,214,215,216,217,218]. In the following parts, we review selected representative studies investigating the associations between serum ferritin levels and the clinical course after allotransplantation.

A previous study compared ferritin levels for 32 AML and 18 ALL patients receiving intensive chemotherapy during a one-year period; the levels were then tested at the time of inclusion and after 3, 6, 9 and 12 months [219]. They observed that a serum ferritin level exceeding 1000 μg/mL was reached at a significantly earlier time point for AML than for ALL patients, i.e., at six months, a higher fraction of AML patients had reached this level. These observations have to be interpreted with great care because the study included relatively few patients for each diagnosis and the antileukemic treatment was not described in detail. However, the observations illustrate that when an unselected group of allotransplant recipients are split into two subsets with and without high pretransplant ferritin levels, respectively, these two patient subsets may show additional and possible differences with regard to diagnosis and pretransplant chemotherapy.

The first retrospective study (Table 6) from the time period 2000–2008 included 99 AML patients and 20 MDS patients; 66 patients were classified as standard-disease-risk (i.e., AML transplanted in first or second remission, having refractory anemia), whereas the others were classified as a high-risk disease (later remission, no remission, refractory anemia with excess of blasts) [207]. Most patients (73%) received GVHD (graft versus host disease) prophylaxis based on tacrolimus and acute GVHD grade II-IV was observed for 48 patients. High ferritin levels were associated with reduced 5-years overall survival and the cumulative risk over relapse was also higher for the high ferritin group. Other factors associated with decreased overall survival were a high disease risk, unrelated donor, use of steroids and acute GVHD grade II-IV. The multivariate analysis showed that the only variables showing an independent association with adverse 5-year overall survival were a high ferritin level and high-risk disease.

The second study (Table 6) included only AML patients [208] and it described significant associations between high pretransplant ferritin and decreased overall survival, as well as decreased relapse-free survival/an increased relapse frequency. This study could not detect any significant association between high ferritin and the risk of GVHD.

The third study (Table 6) was a large retrospective multicenter study including 784 adult patients [209]. The prespecified ferritin cut-off of 2500 μg/L was higher than for the two other studies and these two patient subsets did not show any significant differences with regard to survival or transplant-related mortality. However, the authors defined the optimal thresholds with regard to transplant-related mortality for CRP, albumin and ferritin and based on these levels, they defined three biomarker risk groups that differed with regard to transplant-related mortality. The two parameters CRP and ferritin levels alone showed only a modest but still statistically significant correlation (r=0.35), suggesting that inflammation at least partially is responsible for the increased ferritin.

Several additional retrospective studies including patients with different hematological malignancies have also demonstrated a prognostic impact of iron overload/high ferritin. The main results for eight large representative studies are summarized in Table 7 [210,211,212,213,214,215,216,217] and the overall results summarized in Table 6 and Table 7 suggest that [207,208,209,210,211,212,213,214,215,216,217]:*Strong prognostic impact of pretransplant ferritin:* Associations between high pretransplant ferritin levels and adverse prognosis were observed in all these studies. For example, one study [213] including 590 patients described more than 50% 5-year overall survival for patients with ferritin < 930 μg/L, whereas patients above this cut-off showed 37% survival.*Ferritin is important in AML:* several studies including only AML patients or mainly AML patients together with preleukemic MDS patients have observed an association between adverse prognosis and high pretransplant ferritin levels [207,208,209].*High ferritin has a general effect in allotransplantation:* the association between high ferritin and adverse prognosis has been observed in clinical studies (i) including patients with various hematological malignancies and both intermediate/standard-risk and high-risk disease, (ii) when using both family and matched unrelated donors and (iii) patients receiving various conditioning regimens.*High ferritin associated with increased relapse risk:* six of the nine studies investigating the relapse risk described associations between high ferritin and an increased risk.*High ferritin is associated with increased nonrelapse mortality:* Increased nonrelapse mortality was described for seven of the eight studies summarized in Table 5. There is an increased risk of severe infections [204,207,209,215,217,220]; the causes of death differ between patients [217], but severe GVHD seems less important [218].*The adverse prognosis for ferritin levels exceeding 700–1000 μg/L:* Seven of these eleven studies used a ferritin cut-off between 700 and 1500 μg/L and the most common single cut-off was 1000 μg/L which was used in four of the studies. A significant prognostic impact has not been observed when using a cut-off of 2500 μg/L [203,209].*High ferritin is only partly caused by an increased iron overload:* The pretransplant ferritin levels of 198 de novo AML patients were correlated with the number of previous erythrocyte transfusions; patients with ferritin levels > 1000 μg/L then had received significantly higher numbers of pretransplant erythrocyte transfusions [208]. A similar association was also observed in another study [219]. On the other hand, a meta-analysis showed an adverse prognostic impact by ferritin > 1000 μg/L, but not by an increased iron overload as detected by magnetic resonance imaging.*The acute-phase reaction may contribute to high ferritin:* although the adverse prognostic impact of high pretransplant ferritin seems independent of pretransplant CRP [210] and albumin levels [213], a (possibly minor) contribution of inflammation to the increased ferritin levels seems to be present [209,211].

Increased pretransplant levels are probably caused by a combined effect of iron overload and inflammation. To conclude, high pretransplant ferritin levels seem to have an adverse prognostic impact (i.e., decreased survival) in allotransplant recipients, including AML patients. This decreased survival seems to be caused by complex mechanisms leading to both increased relapse and nonrelapse mortality.

Our conclusions above based on the selected studies summarized in Table 6 and Table 7 are also consistent with the conclusions from the meta-analyses [204,206,211]. Ferritin > 1000 μg/L was commonly used as the cut-off in previous studies [206] and the meta-analyses concluded that there is an association between high pretransplant ferritin levels and decreased overall post-transplant survival [204,206,221], increased nonrelapse mortality [204,206,221], more frequent severe/fungal/blood-stream infections [204,221] and worse progression-free survival [204]. The associations to GVHD seem to be weaker; no association was observed for acute GVHD [204,221], but one study described a significant association with chronic GVHD [204]. Thus, high ferritin levels seem to be associated with death especially due to relapse and severe infections, but less to GVHD [221]. These conclusions regarding allotransplantation in general seem to be true also for AML recipients.

### 7.2. Possible Therapeutic Interventions for Allotransplant Recipients with High Serum Ferritin Levels

Iron chelation has been investigated as a therapeutic possibility in allotransplant recipients with hyperferritinemia. A meta-analysis including 1066 AML patients and 4054 MDS patients described improved overall survival by iron-chelation therapy in MDS, but not in AML patients with ferritin levels exceeding 1000 μg/L, but this treatment did not influence the MDS transformation to AML [205].

A previous study included a consecutive group of allotransplant recipients with AML; a total of 276 had hyperferritinemia (serum ferritin > 1000 μg/L) one month after allotransplantation and 128 of these patients received deferazirox and were compared with the patients not receiving iron chelation [208]. Deferazirox was administered as 10–20 mg/kg/day and continued until ferritin levels were below 500 μg/L. The authors described that those patients receiving deferazirox showed a reduced cumulative incidence of relapse, whereas chronic GVHD was increased. The treatment was also associated with reduced levels of circulating regulatory T cells and sustained high levels of circulating NK cells. Deferazirox dose reduction due to side effects was required in 64% of patients. In our opinion, these observations need to be verified in additional randomized AML studies and the question of deferazirox toxicity has to be better investigated even though the authors of this last study stated that the side effects were manageable.

An alternative strategy is to reduce the risk of pretransplant iron overload and thereby the risk of pretransplant serum ferritin levels > 1000 μg/L. A recent Cochrane analysis concluded that a general transfusion threshold of 7 g/100 mL seems to be safe for most adult patients and this seems to be true also for intensive-care patients [222]. A similar restrictive transfusion strategy was also suggested for hospitalized adult patients with hematologic malignancies, i.e., to consider transfusion when the hemoglobin concentration is below 7 g/dL [223]. Restrictive transfusion strategies in AML are further supported by the observations from recent pilot randomized studies [224,225,226]. Furthermore, a recent retrospective study included 352 AML patients diagnosed between 2007 and 2018 and receiving induction chemotherapy [227]; patients in the less restrictive transfusion group received erythrocyte transfusion when their hemoglobin levels were below 8 g/dL (2007–2014, n = 268), whereas in the restrictive transfusion group, patients received transfusion when levels were below 7 g/dL (2016–2018, n = 84). The less restrictive transfusion group had 1 g/100 mL higher mean hemoglobin levels, received their first transfusions earlier and needed 1.5 more erythrocyte units during their hospital stay following intensive induction chemotherapy. Febrile episodes, CRP levels, admission to the intensive care unit, length of hospital stay as well as response and survival rates did not differ between the two cohorts. The ferritin levels were not registered in this study. In our opinion, the restriction of erythrocyte transfusions at present seems to be the most reasonable strategy to reduce the problem of pretransplant iron overload in allotransplant recipients.

### 7.3. Pretransplant Serum CRP, Albumin and Hepcidin Levels: Associations with Survival, Performance Status, Nutrition and Pretransplant Ferritin Levels

Several studies have investigated the prognostic impact of increased pretransplant CRP levels in patients receiving allogeneic stem cell transplantation for hematological malignancies. A recent meta-analysis was based on 15 previous publications (14 studies and 3458 patients) and the pooled data concluded that increased pretransplant CRP levels were associated with decreased overall survival, an increased risk of nonrelapse mortality and increased risk of acute GVHD [228]. These studies included patients transplanted both with peripheral blood mobilized and bone marrow grafts, and the CRP cut-off was ≤10 mg/L in most studies. One of the studies also described a significant correlation between pretransplant serum CRP and IL6 levels and a significant association between serum IL31 and serum CRP levels [229]. Additional studies have also shown that:A previous study of the pretransplant CRP–albumin ratio in haploidentical allotransplant recipients used a cut-off of 0.087 and a high ratio was significantly associated with lower overall survival also for haploidentical stem cell transplantation [230].Another study suggested that the pretransplant CRP–albumin ratio showed a stronger prognostic impact than the CRP level alone; the impact of this ratio was also observed in the multivariate analysis [231]. Previous studies have described that both CRP and albumin levels are correlated with the ferritin levels and a high ratio is also correlated with several clinical parameters (Table 5). This study showed that a high CRP–albumin ratio was also correlated with a poor performance status.One would expect the albumin levels also to be influenced by the nutritional status. A previous study of allotransplanted AML patients showed that pretransplant weight loss during induction chemotherapy as well as the pretransplant total serum protein level were both associated with reduced overall survival and an increased risk of AML relapse [232]. This prognostic impact was independent of the karyotype and these authors also described decreased leptin levels in patients with pretransplant weight loss. Furthermore, leptin inhibits the proliferation of primary AML cells for a subset of patients [233], but it is not known whether there are altered leptin levels in patients with weight loss or contribute to the increased relapse risk in such patients.Hepcidin levels are also increased during the acute-phase reaction [148] and high pretransplant levels seem to be associated with an increased risk of bacterial and fungal infections post-transplant [180] (see also Section 5.4).

Taken together, these observations suggest that pretransplant signs of an acute-phase reaction have an independent adverse prognostic impact in allotransplant recipients, i.e., decreased overall survival together with increased relapse risk and nonrelapse mortality. The best prognostic parameter seems to be the CRP–albumin ratio. These observations, therefore, support the hypothesis that the adverse prognosis associated with high ferritin levels in allotransplant recipients is possibly caused not only by iron overload, but also an acute-phase reaction.

### 7.4. Altered Nontransferrin-Bound Iron in Allogeneic Stem Cell Transplantation: An Additional Parameter Reflecting Iron Overload and/or Abnormal Iron Metabolism

Labile plasma iron (LPI) is a part of the non-transferrin-bound iron (NTBI); it is redox-active, chelatable and can permeate into organs [234]. Two previous studies suggest that this parameter is altered and has a prognostic impact in AML patients receiving allogeneic stem cell transplantation. First, the ALLIVE study investigated the liver iron content and plasma labile iron level in 112 patients with AML or MDS [235]. The pretransplant evaluation showed no significant correlation between the liver iron content and labile plasma iron. The labile plasma iron increased during conditioning and most patients had increased levels at the day of stem cell transplantation. High pretransplant levels of labile plasma iron were associated with increased non-relapse mortality and a similar prognostic association was also observed for a high liver iron content. Second, the second study compared non-transferrin-bound iron for allotransplant recipients with AML and thalassemia [236]. The pretransplant levels were higher for thalassemia than for AML patients and the levels increased for both groups during conditioning therapy, but AML patients had lower levels both during this pretransplant and during the post-transplant period. The AML patients showed an overlap with healthy controls in their levels of non-transferrin-bound iron, suggesting a heterogeneity of the AML patients.

These observations further illustrate the complexity of the altered iron metabolism in allotransplant recipients and the heterogeneity of these patients. The increased levels during conditioning seem to be related to tissue damage [236]. Ineffective hematopoiesis is a possible cause of abnormal iron metabolism leading to iron overload [148], but it is not known whether or how patients with AML secondary to MDS differ from other AML patients not only with regard to the transfusion history, but also with regard to iron metabolism and the prognostic impact of iron abnormalities in allotransplant recipients.

## 8. The Scientific Basis for Targeting Iron Metabolism/Ferroptosis as a Therapeutic Strategy in Human AML: The Role of Ferritin in Regulation of Ferroptosis

Based in the adverse prognostic impact of both high systemic ferritin levels and the ferroptosis regulatory molecular profile of AML cells, it seems reasonable to hypothesize that therapeutic targeting of iron metabolism/ferroptosis regulation has antileukemic effects in human AML [2,3]. Ferritin is important in both these processes; it is a marker of the iron load, it is involved in cellular uptake and intracellular handling of iron and ferritinophagy is an important step in ferroptosis. The hypothesis is further supported by the observation that AML cells need iron for in vitro proliferation [2,3].

### 8.1. Iron Metabolism in AML Cells: The Importance of Mitochondrial Ferritin

Iron is bound to ferritin as insoluble Fe^3+^; this complex is delivered into cells by receptor-mediated endocytosis and iron is then reduced to the Fe^2+^ form that moves from the endosomes to the intracellular labile iron pool (i.e., iron not bound in a complex) and finally transported to various intracellular destinations (e.g., mitochondria) where it functions as enzymatic cofactors [124,237]. Mitochondrial iron is important for the regulation of cellular metabolism including the tricarboxylic acid cycle, electron transport chain and oxidative phosphorylation [238]. The iron not required for such processes is stored as ferritin-bound redox-inactive Fe^3+^, but the ferritin/iron complex can also be released extracellularly by the iron exporter ferroportin [124,239]. The two main regulators of ferritin synthesis are iron availability and oxidative stress responses [124,238] and intracellular iron overloading with formation of free Fe^2+^ by itself leads to oxidative stress with the formation of free radicals that damage cellular lipids and proteins [124].

Mitochondrial ferritin shows large homology with H-ferritin [240]; its expression is not iron-dependent and it is usually detected only in cells with high metabolic activity [7]. Furthermore, human macrophages reduce their intracellular free iron and increase ferritin expression, including mitochondrial ferritin, during hypoxia [241]. At the same time the ferritinophagy-inducing NCOA4 (nuclear receptor coactivator 4) decreases; this is due to lower *NCOA4* transcription combined with NCOA4 mRNA degradation regulated by mitogen-activated protein kinase 8 (MAPK8/JNK)/signaling. Thus, mitochondrial ferritin and ferritin heavy-chain synthesis seem to cooperate to protect cells from ferroptosis during hypoxia, but this effect was not observed for other non-myeloid cells [241]. A similar hypoxia-induced protection against ferroptosis in myeloid cells may also be operative in the hypoxic bone marrow microenvironment [242] and a recent experimental study concluded that ferritin is an important regulator of mitochondrial iron homeostasis and thereby cellular survival even in AML stem cells [243]. However, additional studies including alternative experimental models are needed to further illuminate the possible importance of mitochondrial ferritin in AML survival/support.

### 8.2. The Cellular Basis for Targeting of Ferroptosis in AML: Regulation of Ferroptotic Cell Death by Iron Metabolism, ROS Production and Lipid Metabolism in AML Cells

Ferroptosis is a form of autophagy-initiated cell death caused by cargo-specific autophagy (i.e., ferritinophagy) [124] and it is, in addition, characterized by oxidative stress and altered lipid metabolism [123,124]. However, other forms of selective autophagy are also involved in the process of ferroptosis, including clockophagy (i.e., the degradation of core circadian proteins [123,244]), mitophagy, lipophagy and chaperon-mediated autophagy [123,124,244]. Ferroptosis is characterized by small abnormal mitochondria with condensed membranes and broken outer membrane, but at the same time, an intact nucleus [123,127]. This form of programmed cell death occurs when three prerequisites are present: (i) specific characteristics/modulation of iron metabolism, (ii) the synthesis of phospholipids containing polyunsaturated fatty acid and (iii) the mitochondrial production of ROS that exceeds the cellular buffering defense [127]. The regulation of all these steps can be altered in AML cells (Table 8). The development of ferroptosis is also influenced/regulated by the amino acid metabolism, intracellular signaling (especially the RAS (RAS proto-oncogene, GTPase), RAF (Raf proto-oncogene, serine/threonine kinase), MEK (Mitogen-activated protein kinase-kinase 7/MEK) and ERK signaling) [127,245]) and oncogenes, including p53 [123,246]. Furthermore, the modulation of the cancer cell microenvironment can alter the lipid metabolism, increase iron accumulation in cancer cells and finally lead to ROS formation [247].

AML cells show increased intracellular iron levels and this seems to be true for cancer stem cells in general that show an increased labile iron pool [159]. The induction of ferroptosis is, therefore, considered as a possible future therapeutic strategy in AML because the cells already have a dysfunctional iron metabolism involving altered levels of iron/ferritin that make them more vulnerable to ROS accumulation. Furthermore, the induction of ferroptosis may contribute to the antileukemic effects of sorafenib and decitabine [127]. However, the antileukemic effect of ferroptosis induction will probably differ between patients because patients are heterogeneous with regard to the AML cell regulation of iron metabolism [164]. For example, a previous study showed that patients with AML cells showing a high constitutive release of proinflammatory cytokines also differ from low-release AML cells with regard to their iron metabolism [4], illustrating that the biological context/cellular microenvironment and AML cell communication with neighboring AML-supporting nonleukemic differs between patients [4,127,255,258].

### 8.3. The Clinical Basis for Targeting of Ferroptosis in AML: The Association Between AML Cell Expression of Ferroptosis-Associated Genes and Patient Survival

Several studies have investigated the possible prognostic impact of the expression of ferroptosis-associated genes in primary AML cells. These studies have investigated gene expression profiles at the mRNA level and taken together, the results suggest that differences in expression profiles of ferroptosis regulators and are associated with survival after intensive chemotherapy:CD8^+^ T cells can mediate anti-cancer activity through the induction of ferroptosis [261].A recent study defined a prognostic signature based on six ferroptosis-associated genes that were expressed in AML bone marrow and the signature was significantly associated with the marrow infiltration of CD8^+^ T cells [262]. This signature could be used to classify AML patients into high- and low-risk subsets and the signature could possibly also be used to refine the ELN risk classification in AML.Another study generated an alternative eight-gene ferroptosis signature that also could be used to stratify patients into high- and low-risk subsets [263].A third study used the expression of only two ferroptosis-associated genes (*DNAJB6* and *HSPB1*) for the prognostication of AML patients [264].Finally, a previous study identified 20 genes whose expression showed statistically significant associations with patient survival [265]. Ferritin light and heavy chains were both included among these 20 initial genes, but not among the final 12 genes used for the construction of their prognostic AML model and classification into either high- or low-risk patient subsets with regard to survival. In vitro studies showed that their high-risk subset was characterized by decreased AML cell susceptibility to several antileukemic drugs. The correlation analyses showed that the identified high-risk AML cells had a higher expression of several immune checkpoint molecules and increased the bone marrow infiltration of M2 macrophages, whereas γδ-T cells were decreased. These results further support the hypothesis that the regulation of iron metabolism is important also for the immune-mediated support of AML leukemogenesis and chemosensitivity.The antileukemic agent erastin increased AML cell line sensitivity to cytarabine and daunorubicin; this drug also induced a mixed pattern of programmed cell death of AML cells with signs of both ferroptosis, apoptosis, necroptosis and autophagic death [266]. It is not known whether this mixed pattern is due to heterogeneity within the hierarchically organized AML cell population with regard to maturity/differentiation or genetic abnormalities.

To conclude, ferroptosis seems to be important for clinical chemosensitivity in AML, i.e., survival after intensive chemotherapy. The modulation of iron metabolisms and/or susceptibility to ferroptosis may also be important for immune-mediated antileukemic reactivity or AML support mediated by immunocompetent cells. However, the possible prognostic impact of ferroptotic AML cell markers needs to be further investigated in prospective clinical studies to clarify whether it is an independent prognostic factor or only a part of a more complex high-risk AML cell phenotype.

## 9. Possible Strategies for Targeting Iron Metabolism or Regulation of Ferroptosis in Human AML

Several therapeutic approaches can be suggested mainly based on experimental studies, but for most of them, the clinical experience evidence is limited or absent. An overview of the various strategies described below is given in Figure 3.

### 9.1. Targeting of Iron Metabolism by Iron Chelation

Iron chelation is a therapeutic strategy that has been tried in allotransplanted AML patients, but rather few studies are available and many of them are relatively small. The observations from selected representative studies are described below.

A large nationwide Korean study included 5395 patients with acute leukemia; the patients were allotransplanted during the period 2003–2015, 65% of the patients had AML, 75% received peripheral blood stem cell grafts and 75% received myeloablative conditioning [218]. The overall early cumulative incidence rate of transplant-related mortality was less than 10% and patients who had received pretransplant iron-chelating therapy had a lower early post-transplant mortality. The study showed that the independent adverse factors for early transplant-related mortality were aged above 40 years, a longer duration from diagnosis to transplantation (median duration 8.8 months for all patients), previous transplantation(s) (6.1% of the patients), cord blood grafts and no pretransplant iron chelation (*p*-value at 50 and 100 days being *p* < 0.001). This large study thus suggests that pretransplant iron overload due to erythrocyte transfusions had a prognostic impact in these AML patients. However, it should be emphasized that the interval from diagnosis to transplantation was relatively long compared with the present practice.Another study investigated the use of iron chelation immediately before stem cell transplantation during myeloablative busulfan-based conditioning therapy and this treatment with deferazirox reduced the labile plasma iron without causing severe toxicity in any of the 25 patients [267].Other studies have investigated deferazirox iron chelation later after hematopoietic reconstitution for allotransplant recipients with signs of iron overload [268,269,270,271]. All these studies concluded that the iron chelation could reduce iron overload without having severe toxicity.

Taken together, these studies show that it may be possible to use iron chelation as a therapeutic alternative to venesectio for the treatment of post-transplant iron overload. Experimental in vitro studies suggest that the iron chelator desferroxamin can also alter the constitutive cytokine release by primary human AML cells [4] and thereby influences both autocrine signaling loops as well as the communication with neighboring cells in the common bone marrow microenvironment. However, additional studies are needed to clarify whether such direct effects on the AML cells have any clinical relevance.

### 9.2. Possible Strategies for Modulation of Iron Metabolism: Are They Relevant for AML?

The transferrin receptor is expressed by primary AML cells. The levels are usually lower than for normal CD34^+^ bone marrow cells, but a subset of patients show a high expression level, similar to the corresponding normal cells [147]. Both in vitro studies and animal models have demonstrated that transferrin receptor antibodies can inhibit AML cell proliferation [4,272,273,274,275]. This therapeutic strategy seems to cause the upregulation of cell surface transferrin receptor levels (reduced internalization?) with a decreased transferrin uptake [274], an antiproliferative effect without accumulation at a certain step in the cell cycle [272,273,274,275] and a modulation of the constitutive cytokine release with increased extracellular levels of both chemokines, interleukins and metalloproteases [4]. The last observation suggests that there is a link between iron metabolism and communication between AML cells and neighboring AML-supporting cells. Even though one would expect such targeting also to influence normal hematopoiesis, a clinical study of the 42/6 receptor antibody in cancer patients showed that systemic iron levels were reduced, but at the same time, the toxicity seemed acceptable [276]. However, clinical studies in AML are, to the best of our knowledge, not available.

Synthetic hepcidin (see Section 5.4) mimetics are now available [148]; they seem to reduce systemic iron levels and be well-tolerated based on observations in phase I clinical studies [277,278]. A hepcidin monoclonal antibody has also been tested in early clinical studies; it increased serum iron as well as transferrin saturation and thus seems to cause iron mobilization [279]. Alternatively, the transmembrane protease matripase 2 (MT2A) is predominantly expressed in hepatocytes and suppresses hepcidin expression by cleaving the hepcidin regulator hemojuvelin into an inactive form [280,281]. An alternative strategy for targeting the hepcidin–ferroportin axis is, therefore, a specific inhibition of MT2A expression or its protease activity [280,281,282,283,284], targeting of bone morphogenic protein 6 that inhibits hepcidin expression [285] or direct targeting of ferroportin by monoclonal antibodies [286]. Although some of these strategies have been investigated in early clinical investigations, results from AML studies are not available and it is, therefore, not known whether the targeting of hepcidin will have any beneficial effects in AML.

### 9.3. Modulation of Iron Metabolism and Regulation of Ferroptosis in AML Cells by Histone Deacetylase Inhibitors: Increased Intracellular Labile Iron Pool

A recent study investigated the effects of the histone deacetylase (HDAC) inhibitors entinostat and vorinostat in AML cells and they observed that HDAC inhibitors upregulated the expression of genes that increased the intracellular labile iron pool [287]. The altered expression of the ferritin-degrading mediator NCOA4 (nuclear receptor coactivator 4) and the heme-degrading HMOX1/2 (heme oxygenase 1/2) were important for the sensitization to ferroptosis caused by these two inhibitors.

### 9.4. AML Targeting by Iron Oxide Nanoparticles: Induction of Ferroptosis in Response to Increased ROS Levels

The FDA-approved iron oxide nanoparticle ferumoxytol (Feraheme) can be used for the treatment of an iron deficiency, but it can also be used to increase intracellular iron of AML cells when the leukemic cells express low levels of the iron exporter SLC40A1 (solute carrier family 40 member 1/ferroportin) [239]. Increased intracellular free ferrous iron levels then lead to reactive oxygen species (ROS) production, increased oxidative stress and, finally, cell death. Studies in animal models, including patient-derived xenografts, suggest that this nanoparticle can mediate antileukemic effects [239].

Most AML patients have leukemic cells that show high constitutive production and a release of ROS due to the constitutive activation of NOX (nicotine amide adenine dinucleotide phosphate oxidase) with production of H_2_O_2_, a mild pro-oxydant molecule that can act as a second messenger and promote AML cell proliferation [288]. The increased ROS levels are associated with decreased glutathione levels and decreased p38^MAPK^ responses [288]. On the other hand, iron oxide nanoparticles can increase intracellular ferrous iron and convert H_2_O_2_ to the highly reactive hydroxyl radical OH^−^; this last mediator cleaves heat-shock protein 90 as an initial step that finally leads to the dysfunction of the G2/M checkpoint and ROS-dependent cell death [289,290]. Iron oxide nanoparticles can thereby increase the AML cell susceptibility to other antileukemic agents, e.g., increased susceptibility of monocytic ROS-high AML cells to the BCL2-inhibitor venetoclax [290] and the conventional cytotoxic drug cytarabine [289].

### 9.5. Other Possible Strategies for Targeting of Iron Metabolism or Induction of Ferroptosis in AML

The general strategies that have been considered for induction of ferroptosis include (i) the modulation of iron metabolism with cellular accumulation of iron, (ii) induction of ferritinophagy, (ii) inducing accumulation of polyunsaturated fatty acid-phospholipids or (iv) depletion/inhibition of natural inhibitors of ferroptosis [291]:*Targeting of AMPK:* Dihydroartesimin induces the induction of AMPK phosphorylation and thereby the inhibition of mTOR/p70S6k signaling in AML cells; these events induce autophagy, increase ferritin degradation, increase the unstable iron pool and ROS accumulation and, finally, cause ferroptosis [292,293]. Furthermore, the pollen-derived flavonoid Typhaneoside increases intracellular and mitochondrial ROS levels and it seems to have an antiproliferative effect in AML cells that is mediated through AMPK activation followed by triggering of autophagy, ferritinophagy/ferritin degradation, ROS accumulation and ferroptosis [294]. Finally, animal studies suggest that this therapeutic strategy has a limited general toxicity [294].*NFκB targeting:* The NF-E2-related factor 2 (NRF2) shows increased NFκB-driven constitutive expression in primary AML cells compared with normal CD34^+^ bone marrow cells [295,296]. NRF2 expression is particularly high in AML cells with genetic abnormalities associated with adverse prognosis [297]. Several downstream NFR2 targets are directly involved in the regulation of ferroptosis, including glutathione peroxidase 2 (GPX4) that is a key regulator of ferroptosis, is upregulated in human AML and particularly high levels are associated with adverse prognosis [298,299]. NRF2 is thereby a regulator of antioxidant responses and is thus important for AML cell survival [295,296,297,298,299]. High NRF2 expression is associated with resistance to both conventional cytotoxic drugs (e.g., cytarabine) [296,297] and the BCL2 inhibitor venetoclax [298]. Finally, the antileukemic effect of NFκB inhibition may, therefore, at least partly be mediated by antagonizing this NRF2 effect [296,300].*Retinoids:* The all-trans retinoic acid derivative ATPR can inhibit NRF2; this agent is known to have antileukemic effects through increased lipid peroxidation and increased lipid ROS production [295]. The effect of ATPR on iron metabolism was also associated with monocytic differentiation [295]; the sensitivity to oxidative stress could thereby be increased and autophagy was promoted [301]. ATPR also increases ROS levels and this is possibly due to ferritin degradation and/or modulation of other components of the cellular iron metabolism [302].*Fatty acid metabolism:* A recent study described altered fatty acid metabolism during ferroptosis in human AML cell lines [303]. Twelve fatty acids were significantly altered in AML cells during ferroptosis, including dihomo-γ-linoleic acid, arachidonic acid and docosahexaenoic acid. Exposure to exogenous dihomo-γ-linoleic acid could then induce ferroptosis and this proferroptotic effect was dependent on the enzyme acyl-CoA synthetase family member 4. Taken together, these observations suggest that targeting fatty acid metabolism should be further explored as a possible antileukemic and proferroptotic strategy in AML.*Lipid peroxidation:* Imetelstat is a telomerase inhibitor that mediates antileukemic effects in patient xenograft models by proferroptotic effects [304]. This antileukemic activity is seen especially in AML cells with mutant NRAS and oxidative stress-associated gene expression signatures and this activity was mediated through the increased formation of polyunsaturated fatty acid-containing phospholipids leading to increased levels of lipid peroxidation and thereby oxidative stress [304]. This example also supports the hypothesis that targeting fatty acid metabolism can have a proferroptotic effect.*Glutathione inhibition:* This strategy can induce ferroptosis in AML cells through the induction of lipid peroxidation [304,305]. The inhibition of glutathione peroxidase 4 (GPX4, see the NFR2 chapter above) can induce ferroptosis in AML cell lines with the characteristic mitochondrial lipid peroxidation, and additional degradation of the mitochondrial electron transport chain enhanced the antileukemic effect of GPX4 inhibition [306]. Mitochondrial functions including energy metabolism are thus important/involved in the regulation of ferroptosis. Finally, inhibition of GPX4 can induce ferroptosis, and inhibition of GPX4 together with inhibition of the upstream NRF2 have synergistic effects [299].*Erastin:* The agent erastin seems to increase AML cell line sensitivity to cytarabine and daunorubicin, and this agent induces a mixed pattern of programmed cell death in AML cells with signs of ferroptosis as well as other forms of programmed cell death [266]. The molecular mechanisms behind this erastin effects are not known, but may involve activation of c-JUN N-terminal kinase and p38.*Intracellular signaling:* Various intracellular signaling pathways are important for the induction of ferroptosis, including the RAS-MAPK8(JNK)/P38 pathway in erastin-induced ferroptosis [307]. Erastin is then able to increase the susceptibility to cytarabine and doxorubicine. The ferroptosis-inducing effect of AMPK activation suggests that the status of the PI3K-Akt-mTOR pathway is also important for induction of ferroptosis [308,309].*Honokiol*: This is a natural small molecule that can induce ferroptosis in AML cells through the increased expression of the Heme oxygenase 1 enzyme [310]. However, this agent seems to have multiple effects in AML cells, including the modulation of STAT3 signaling, induction of proteasomal protein degradation, modulation of gene expression and causing cell-cycle arrest [311,312,313].

It should be emphasized that these possible strategies are suggested based on experiments and, for many of them, they are mainly or only ex vivo observations. More detailed studies of primary patient-derived AML cells are needed, the question of risk of severe toxicity should be addressed especially by investigating effects on normal hematopoietic cells, and the question of patient heterogeneity has to be addressed (see Section 9.7).

### 9.6. Targeting of the Chemokine Network

Macrophages with the M2 phenotype release the CCL20 chemokine and this chemokine seems to enable AML cells to maintain their iron hemostasis by the upregulation of SLC7A11, mitigation of mitochondrial damage and inhibition of ferroptosis [314]. This chemokine can also be constitutively released by the AML cells for most patients [315] and CCL20 as a mechanism of chemoresistance may, therefore, involve both autocrine and paracrine loops.

These chemokine-mediated effects may be relevant for the resistance to the cytotoxic agent daunorubicin (an anthracycline) that increases reactive oxygen species and lipid peroxidation of AML cells, and these initial events are accompanied by SLC7A11/GPX4 inhibition together with increased intracellular free iron with mitochondrial damage and induction of programmed cell death [314]. The effects of CCL20 will antagonize these daunorubicin effects, and targeting CCL20 may thereby increase the susceptibility of AML cells to this conventional cytotoxic drug.

### 9.7. The Question of Patient Heterogeneisty and the Heterogeneity of the Hierarchically Organized AML Cell Population

AML patients are heterogeneous with regard to iron metabolism and the regulation of ferroptosis in their leukemic cells. This heterogeneity includes several molecular mechanisms [147,316]:Patients are heterogeneous with regard to ferritin levels as well as expression of the two iron-regulatory genes *ACO1* and *IREB2* in their leukemic cells [316].AML cells that show similarities with normal immature hematopoietic cells with regard to iron/ferritin internalization and intracellular release have an adverse prognosis [147]. Such biological characteristics are present especially in a minority of patients with few signs of differentiation [147].AMPK is an important regulator of PI3K signaling through its modulation of the downstream AKT-mTOR activation and thereby also the regulation of autophagy and ferroptosis [292,293], but the constitutive as well as the insulin-induced activation of the PI3K-AKT-mTOR pathway differs between AML patients [317].AML patients are heterogeneous with regard to pretreatment hyperferritinemia at the time of first diagnosis (see Section 5.1 and Section 5.3) and the development of iron overload during the initial induction/consolidation treatment [192].AML patients are also heterogeneous with regard to pretreatment hepcidin levels, and only a subset of patients show increased systemic levels compared with healthy controls [178,179,180].AML patients are also heterogeneous with regard to the plasma labile iron pool [235,236].

Most of the possible strategies described above for targeting iron metabolism/ferroptosis have only or mainly been investigated in experimental models, the only strategies examined in clinical studies are HDAC inhibition [318,319]. and iron chelation (Section 9.1). Additional studies, therefore, have to clarify which of these strategies that are effective and have an acceptable toxicity in AML, and these studies also have to focus on patient heterogeneity and whether certain strategies should be preferred in biologically defined AML subsets.

Another question is the heterogeneity of the hierarchical AML cell population. A recent study described the antileukemic effects of targeting NCOA4 (a chaperone important for ferritinophagy) [287,320]. Quiescent AML stem cells are possibly of particular importance for the development of later chemoresistant AML relapse; these cells have a distinct gene expression signature, show increased autophagy and differ in their iron metabolism when compared with cycling cells [320]. These cells depend on ferritinophagy that is mediated by NCOA4 (see Section 9.3) and the suppression of NCOA4 was particularly toxic for the CD34^+^CD38^−^ AML cell population, indicating a specific vulnerability of these cells to ferritinophagy disruption. Thus, the effect of targeting iron metabolism/ferroptosis may differ between AML cell subsets, and effects on quiescent AML stem cells may then be of particular importance.

## 10. Discussion

### 10.1. Ferritin in Human AML

The previous sections clearly show that ferritin has multiple and complex biological functions in AML. Ferritin is not only an adverse prognostic biomarker in AML, but also an important regulator of AML cell survival/chemosensitivity and its effects on AML cells are probably mediated both by direct effects on the AML cells and indirect effects via neighboring nonleukemic cells in the bone marrow microenvironment.

Hyperferritinemia can have many different causes and requires a systematic diagnostic strategy outline in several recent publications [321,322]; this is also true for AML patients hyperferritinemia.

Uncommon cases of spontaneous AML remission have been described, including remission following systemic bacterial infection [323,324,325]. The biological mechanisms behind such remissions are not known, but one would expect that the infections in such patients are associated with signs of inflammation, including a strong acute-phase reaction. However, signs of inflammation/acute-phase reaction are generally associated with chemoresistance/AML support (see Section 5), and infections are common but spontaneous remissions are very uncommon in human AML. Thus, if inflammation contributes to development of spontaneous AML remission this will probably occur only for patients with a specific and uncommon AML cell genotype/phenotype or development of an inflammatory response with a specific phenotype.

The results described in this review suggest that ferritin has several important clinical and biological functions that may contribute to and explain the association between high ferritin and adverse prognosis in human AML:Both ferritin and other acute-phase biomarkers are associated with prognosis/survival in AML; this is true both for ferritin levels at the time of first diagnosis and the levels before allogeneic stem cell transplantation. The acute-phase reaction will, therefore, represent a biological context for the ferritin-associated prognostic impact in many patients, but despite the association between ferritin and other acute-phase markers several studies have observed an independent prognostic impact of ferritin.Ferritin can reflect iron overload, and this may then represent an additional biological mechanism behind the prognostic impact in allotransplant recipients compared with newly diagnosed patients. The iron overload may influence the iron metabolism in the AML cells, bone marrow stromal cells and/or immunocompetent cells.Direct ferritin effects on the AML cells seem to involve regulation of iron metabolism, intracellular signaling and regulation of cell survival/programmed cell death (Section 2.2).Indirect ferritin effects mediated via neighboring cells in the AML-supporting bone marrow microenvironment may include induction of M2-polarization of macrophages or modulation of endothelial cells/angiogenesis (Section 3.2 and Section 3.3).Experimental studies suggest that ferritin has immunosuppressive effects that involve both the innate immunity by suppression of myelopoiesis (neutrophils, monocytes, dendritic cells) and the adaptive immune system through direct and indirect effects on the function of and balance between various T cell subsets (Section 3.2). Such effects may influence both antileukemic immune reactivity (e.g., after allogeneic stem cell transplantation) and AML supporting immunocompetent cells.Ferritin may be important for AML cell susceptibility to targeted therapies through its role in regulation of different steps in ferroptotic programmed cell death.

Many of the cellular ferritin effects have only/mainly been characterized in experimental models, but the adverse prognostic impact of high systemic ferritin levels has been demonstrated in several clinical studies. The experimental studies suggest that the biological/molecular mechanisms behind this prognostic impact are highly complex and need to be characterized in detail. The question of patient heterogeneity with regard to iron metabolism of AML cells also requires further studies.

The main impact/associations of ferritin in allotransplant recipients is decreased overall survival due to increased relapse risk and increased transplant-related mortality including severe infections, whereas the association with GVHD is weaker/absent and thus seems to be less important. A possible hypothesis is that ferritin/iron have general immunomodulatory effects finally leading to weaker graft versus leukemia reactivity and a weaker defense against infections, but at the same time also weaker graft versus host reactivity. Effects on vasculature/endothelial cells may also contribute to this because vascular modulation is important both for chemotaxis/immunoregulation and for the development of several post-transplant complications (Section 3.2 and Section 3.3) [129,326,327,328,329].

### 10.2. What Is the Optimal Acute-Phase Biomarker to Be Used in Human AML?

The detection of an acute-phase reaction in AML seems to have a prognostic impact, but in our opinion several acute-phase biomarker are not optimal for use in AML patients. Normal peripheral blood cell counts are strongly influenced by the bone marrow infiltration of AML cells and less by acute-phase reactions, albumin is influenced by the nutritional status and fibrinogen by the coagulation status [131,138,199]. In our opinion CRP is the best biomarker to use because the influence of inflammaging is also an effect of inflammation, but it should possibly be used in combination with ferritin because this marker seems to reflect a specific inflammatory phenotype associated with adverse prognosis [199].

### 10.3. Therapeutic Targeting of Iron Metabolism/Ferroptosis: Toxicity Versus Efficiency

Several possible therapeutic strategies for targeting of iron metabolism/ferroptosis have bee suggested (see Section 9). However, several of these strategies are suggested mainly or only based on experimental, and their toxicity in AML or in combination with established AML therapy remains to be established. Possible strategies with known toxicity profiles are iron chelation (including the newer chelators), HDAC inhibitors, retinoids and PI3K-Akt-mTOR inhibition [218,287,292,293,302,318,330,331,332].

### 10.4. Ferritin for Cargo Delivery

The ferritin molecule has a 24-subunit cage-like structure with an internal core, and ferritin nanoparticles with a similar architecture has now been constructed. The advantage with these particles is that they are small, probably has good biosafety and low risk of initiating immune responses [333,334,335,336]. Ferritin may, therefore, be an alternative to other naturally designed structures (e.g., exosomes) for specific cargo delivery [335]. Several strategies for the clinical use of ferritin particles have been reviewed previously [336]. First, one alternative is to use such particles to encapsulate metal ions and thereby create imaging probes for visualization of tumors that often show high expression of ferritin-binding iron-internalizing receptors. Second, ferritin nanoparticles may be used for targeted delivery of drugs, e.g., apoferritin-encapsulated particles containing doxorubicin and cisplatin have been constructed. Third, another strategy is to load the external surface of such ferritin molecules with protein drugs or with specific molecules directed against cancer-expressed surface molecules. Finally, an alternative strategy is to target therapeutic agents to receptor-expressing cells by binding transferrin or antibodies specific for the transferrin receptor to the surface. All these diagnostic/therapeutic strategies are based on the observation that malignant cells often show high levels of transferrin/ferritin-binding receptors. The construction of such nanoparticles can be based both on binding of H- and L ferritin [334,336]. AML is also characterized by altered cellular iron metabolism, but it should be emphasized that to the best of our knowledge none of these strategies have been investigated in AML. One would also expect receptor-directed therapies to affect normal cells, and the question of toxicity therefore has to be carefully addressed.

## 11. Conclusions

The systemic ferritin level is an adverse prognostic biomarker in human AML. These prognostic associations are probably caused by complex effects on both leukemic and leukemia-supporting non-leukemic cells. The biological mechanisms behind this prognostic impact probably vary between patients. A more detailed characterization of these biological mechanisms may identify new and more specific prognostic markers and possibly also allow individualized therapeutic targeting of iron metabolism/ferroptosis regulation.

## Figures and Tables

**Figure 1 ijms-26-05744-f001:**
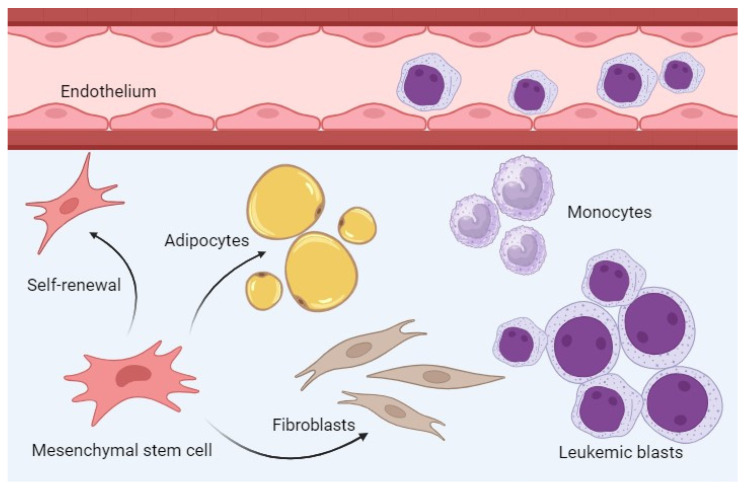
The bone marrow microenvironment in human AML; non-leukemic cells that (i) communicate with hierarchically organized AML cells and (ii) are influenced by soluble extracellular ferritin. Transendothelial AML cell migration with peripheral blood leukemization is seen for many patients, whereas extramedullary infiltration is uncommon. Soluble extracellular ferritin can affect at least endothelial cells, mesenchymal stem cells, fibroblast subsets and adipocytes as well as monocytes/macrophages. Possible ferritin effects on other AML-supporting cells (see the text) have not been characterized, and these cells are not shown in the figure.

**Figure 2 ijms-26-05744-f002:**
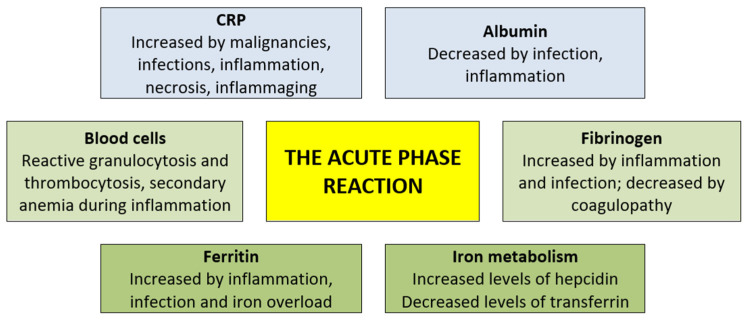
The effects of the acute-phase reaction on iron metabolism are complex; ferritin levels increase whereas, at the same time, transferrin levels are decreased [110]. The serum levels of iron are, in addition, decreased and this seems to be at least partly due to the accumulation of iron in macrophages [110]. Furthermore, the effect of inflammation/an acute-phase reaction on the levels of circulating normal myeloid cells are divergent, neutrophils are increased, thrombocytosis is also common, and many patients have anemia [110]. Finally, the function of the hepatocytes that express/release many of the acute-phase proteins is altered and there are additional systemic neuroendocrine changes (e.g., increased catecholamines, corticotropin and cortisone), several metabolic changes (e.g., a loss of muscle and negative nitrogen balance, increased vasopressin, decreased IGF-1, increased hepatic lipogenesis, but increased lipolysis in adipose tissue) and changes in not-protein plasma constituents (e.g., hypozinchemia, hypercupremia and increased glutathione) [110,111]. The figure presents protein biomarkers of inflammation (blue), normal peripheral blood cells (including platelets)/coagulation (light green) and biomarkers of iron metabolism (dark green).

**Figure 3 ijms-26-05744-f003:**
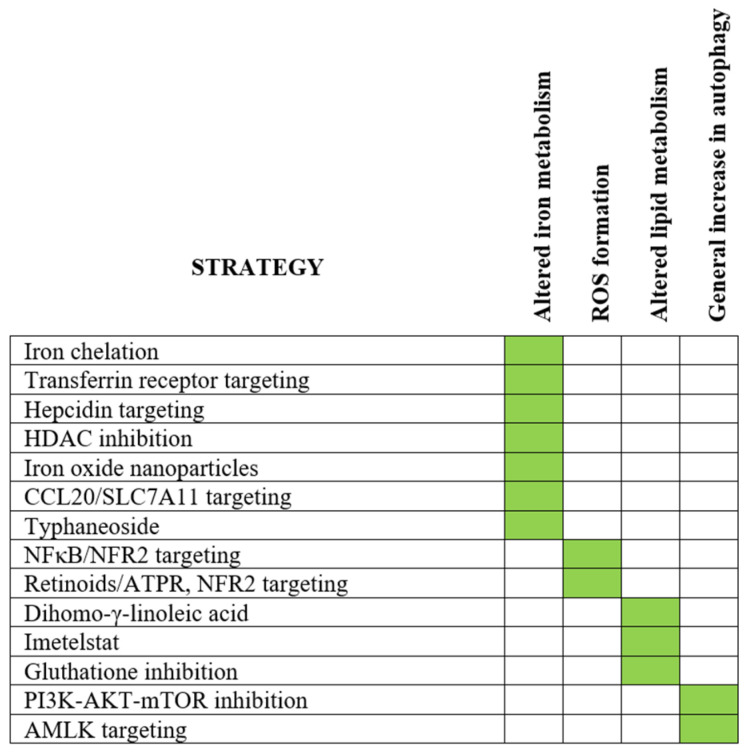
An overview of possible therapeutic strategies for targeting iron metabolism and/or induction of ferroptosis. The strategies are described more in detail in the text. Induction of ferroptosis includes the three main events: (i) altered iron metabolism, (ii) induction of oxidative stress and (iii) altered lipid metabolism (see Table 8) and, in addition, we indicate whether the main mechanism was a general increase in autophagy. The figure shows which of these three events that represent the main target for each of the described strategies (for references, please see the text in Section 9).

**Table 1 ijms-26-05744-t001:** A summary of the fundamental biological characteristics of ferritin [5,6,7,8,9,10,11,12,13,14,15,16,17,18,19,20,21,22,23,24,25,26,27,28]. For more detailed descriptions, see Section 2.1.

Molecular Structure	Cellular Expression	Distribution
**Ferritin chains/subunits:** Ferritin light chain (FRL; weight 10 kDa). Ferritin heavy chain (FRH; weight 21 kDa). 55% homology between the chains Similar three-dimensional structure. **Encoding genes:** Ferritin light-chain chromosome 11q Ferritin heavy-chain chromosome 19q **Molecular structure of ferritin:** A spherical macromolecule formed by 24 subunits. The inner iron-containing nanocage has six hydrophobic and eight hydrophilic channels: the last ones are the main pores for iron uptake.	**Increased transcription:** (i) proinflammatory cytokines (IL1β, IL6, TNFα) via NFκB, (ii) IFNγ and TLR4 ligation through nitric oxide-dependent mechanisms involving IRP2 degradation. Most important for FTH expression. **Post-transcriptional regulation:** Mediated through the iron-responsive elements of ferritin-encoding RNAs; adjustment of ferritin level to intracellular iron level. More important for FTL than for FTH. **Ferritinophagy:** Degradation by autophagy, a part of programmed ferroptotic cell death.	**Intracellular compartmentalization:** 2:1 distribution between membranous compartments and cytosol. **Intracellular functions** Iron oxidation and storage, an antioxidant effect by reducing toxic free iron, regulation of iron metabolism and ferroptosis. **Extracellular release/distribution:** Secreted via the multivesicular body–exosome pathway and autophagosome-related pathways. *Serum protein binding:* high molecular weight kininogen, apolipoprotein B, α-2-macroglobulin and fibrinogen. *Receptors:* CD71, CD204, Scavenger receptors, TIM-2, CXCR4 (for details see Section 2.2).

Abbreviations: FTH, ferritin heavy chain; FTL, ferritin light chain; IFN, interferon; IL1β, Interleukin 1β; IRP2, iron-responsive element-binding protein 2; NFκB, Nuclear factor kappa-light-chain enhancer of activated B cells; TLR4, Toll-like receptor 4; TNFα, Tumor necrosis factor α.

**Table 2 ijms-26-05744-t002:** Effects of iron and iron overload on MSCs; a summary of important effect described for human MSC and/or in various animal models [100,101,102,103,104,105,106,107].

**Cellular ferritin uptake**
Iron enters MSCs both through transferrin-dependent and -independent mechanisms [102].
**Proliferation**
Decreased MSC proliferation in iron overload has been described [101,103], but another in vitro study described increased proliferation of in vitro-cultured MSCs exposed to iron concentrations corresponding to systemic levels observed in patients with iron overload [102]. Altered cell cycle regulation [101,102].
**Cell survival**
Increased apoptosis has been described during iron overload [101]; it can be accompanied by mitochondrial fragmentation due to high ROS levels and enhanced autophagy during iron overload [104]. ATP concentrations were also decreased due to high ROS levels and reduced respiratory chain activity. Similar observations were made both for normal murine MSCs and MSCs derived from MDS patients.
**Intracellular signaling**
Increased mRNA and protein expression of PI3K and FOXO3 (Forkhead box protein O3) [103]. Increased AMPK activation leading to mitochondrial fragmentation [104]; increased ERK1/2 activation, but possibly decreased PI3K-AKT signaling [100]. Activation of MAP kinase pathways [102] and reduced CCR2 expression [100] have also been described.
**Cellular metabolism**
Increased ROS production [101]. Decreased ATP production and decreased respiratory chain activity [104]. Iron can induce upregulation of ferritin expression [105].
**Differentiation**
Inhibition of osteoblastic differentiation [102], whereas effects on adipogenic and chondrogenic differentiation are absent/weaker [105]. Ferritin treatment can have an anti-osteogenic effect [105].
**Soluble mediator release**
Bone marrow MSCs show decreased levels of IL10, CXCL12, IGF1, SCF, VEGF and MMP2/9 [103,106].
**Effects of iron overload on normal hematopoiesis**
Murine studies suggest an increased number of myeloid progenitors, whereas the number and function of erythroid progenitors and hematopoietic stem cells were not altered [101,106]. Bone marrow transplantation to recipients with iron overload was then associated with delayed hematological reconstitution. The bone marrow is characterized by increased oxidative stress [106]. Reduced support of normal hematopoiesis [103].
**Systemic effects**
Mice with iron overload show increased systemic levels of proinflammatory TNFα and IL6 together with increased systemic ROS levels and altered bone microarchitecture [107].

**Table 3 ijms-26-05744-t003:** Causes of hyperferritinemia defined as serum ferritin exceeding 5000 μg/L [118]. The results are presented as percentage of the total number of cases. Many patients had more than one cause.

**Infectious diseases 39%**
Bacterial 15%, fungal 5%, viral 4%, others 1% and agent not documented 14%.
**Hemophagocytic lymphohistiocytosis (HLH) 19%**
Together with hematological malignancies 13%, infections 6% (including 4% with viral infections), other uncommon causes of HLH including solid cancers, primary HLH, rheumatic/inflammatory disease and drugs (1.2%).
**Hepatitis 15% (defined as aminotransferase levels ≥ 10 times upper normal limit)**
Cancers and infections were most common; also including drug-induced, alcoholic and rheumatic/inflammatory disease.
**Iron overload syndrome 13%**
Cancers 11%, hemoglobinopathy 2%
**Malignancies 6%**
Hematological malignancies and solid cancers, 3% each
**Others 8%**
Cytokine release syndrome (only diagnosed for Car-T cell recipients) 3%, rheumatic/inflammatory disease 2%, acute hemolysis 1.4% and not classified 1.8%

**Table 4 ijms-26-05744-t004:** Effect of increased pretreatment/baseline acute-phase biomarkers at the time of first AML diagnosis (non-APL variants); a summary of the results for studies investigating other biomarkers than systemic ferritin levels. All studies were retrospective. A statistically significant increase or decrease in overall survival, event-free survival or relapse rate is indicated by arrows (↑↓).

Patients [Reference]	Parameter (Cut-Off)	Overall Survival	Event-Free Survival	Nonrelapse Mortality	Comment
206 patients, median age 54 years (range 17–74 years) [161]	CRP (>150 mg/L)			↑	Increased 60 days induction mortality in multivariate analysis
243 patients, median age 47 years (range 14–80 years) [162]	Albumin (<35 g/L)	↓	↓		Decreased survival in multivariate analysis
756 patients, median age 60 years (range 18–85 years) [163]	Albumin (<25, 25–35 and >35 mg/L)	↓		↑	Increased 30 and 60 days mortality; increased grade ≥ 3 toxicity
212 patients, median age 49 years (range 7–82 years) [164]	CRP:albumin ratio (1.015)	↓	↓		Ratio correlated with ferritin level. Ratio better predictor of prognosis than CRP, albumin or ferritin as single markers
188 patients, age ≥ 65 years, transplant-ineligible [165]	CRP:albumin ratio	↓			Decreased survival for favorable, intermediate and adverse genetic risk groups
215 patients, median age 40 years (range 14.65 years) [166]	Fibrinogen (>3.775 g/L)	↓	↓		
375 patients, median age 53 years (range 17–77 years) [167]	Fibrinogen (>4.1 g/L)	↓	↓		
282 patients, median age 51 years (range 19–66 years) [168]	FAC ratio ^1^ Cut-off 3.06	↓	↓		Over mortality and mortality after 6 months increased
328 patients, median age 49.5 years (range 15–75 years) [169]	FAC ratio ^1^ Cut-off 1.44	↓	↓		The prognostic impact was seen both for patients with intermediate and adverse ELN risk classification

^1^ FAC ratio was defined as [fibrinogen [(g/L) × CRP (mg/L)]: albumin (g/L).

**Table 5 ijms-26-05744-t005:** The pretreatment acute-phase reaction in patients with newly diagnosed AML (non-APL variants), a summary of observations from available retrospective clinical studies [154,156,157,161,162,163,164,165,166,167,168,169]. The table shows associations between (i) various acute-phase parameters and clinicobiological AML cell characteristics and (ii) associations between the different acute-phase markers/proteins. Significant associations/correlatios are marked with +, no association with − and associations not tested are marked as not tested/nt.

Study	Acute-Phase Parameter (Cut-Off)	Age	Secondary AML	Leukemization; Increased WBC Count the Blood	WBC Count/Bone Marrow Blasts (%)	FAB Classification	Cytogenetic Risk Classification	Remission After 1/2 Cycles		Ferritin Level		CRP Level	Albumin Level	Fibrinogen Level
[154]	Ferritin (>750 μg/L)	−	−	nt	nt	nt	−	−				−	nt	nt
[156]	Ferritin (>400 μg/L)	−	−	+	−	−	+	nt				+	+	nt
[157]	Ferritin (>4 × upper normal limit)	−	nt	+	nt	−	−	−				+	nt	nt
[161]	CRP (>150 mg/L)	+	nt	nt	nt	+	+	nt		nt			nt	nt
[162]	Albumin (<35 g/L)	+	nt	+	+	−	−	+		nt		nt	nt	nt
[163]	Albumin (<25, 25–35 and >35 g/L)	+	+	nt	nt	nt	−	nt		nt		nt		nt
[164]	CRP–albumin ratio (1.015)	+	+	−	+	+	+	+		+		+	+	nt
[165]	CRP–albumin ratio	nt	nt	nt	nt	nt	nt	nt		nt		nt	nt	nt
[166]	Fibrinogen (3.775 g/L)	−	nt	−	−	−	−	−		−		−	−	
[167] ^1^	Fibrinogen (4.1 g/L)	−	−	−	−	+	+	nt		nt		+	nt	
[168]	FAC ratio (3.06) ^2^	−	nt	+	+	−	nt	nt		nt		nt	nt	nt
[169]	FAC ratio (1.44) ^2^	−	nt	+	+	nt	+	nt		nt		+	+	+

^1^ Including 31 patients with acute promyelocytic leukemia. ^2^ Fibrinogen-CRP–albumin ratio.

**Table 6 ijms-26-05744-t006:** The associations between transfusion load/serum ferritin and survival after allogeneic stem cell transplantation; a summary of the studies including patients diagnosed with AML and MDS. These studies included only or mainly patients with AML or preleukemic MDS, but they differed with regard to the cut-off level defining patients with high and low pretransplant ferritin level.

**Study [Reference]**	[207]	[208]	[209]
**Time period (years)**	2000–2008	2007–2012	2008–2010
**Patient number**	119	198	784
**Median age (range)**	41 years (18–63 years)	41 years (19–66 years)	50 years (18–78 years)
**Diagnosis (number)**	AML (99), MDS (20)	All de novo AML	AML 626, MDS 129
**Related donor (number)**	Related donor 54	Sibling donor 179	-
**Marrow grafts (number)**	74	144	136
**Conditioning (number)** **Myeloablative/others**	92/27	143/55	566/218
**TBI**	-	-	177
**Pretransplant erythrocyte transfusions (number)**	-	Low ferritin 13 (0–32) High 19 (0–56)	-
**Median ferritin (range)**	972 μg/L (31–11 500 μg/L)	1075 μg/L (10–7455 μg/L)	1148 μg/L (51–14,298 μg/L)
**Cut-off level for ferritin**	1000 μg/L	1000 μg/L	2500 μg/L
**Overall disease-free survival High/low**	5-years overall survival 30%/70%	49%/72%	No significant association
**Relapse rate or relapse-free survival; high versus low pretransplant ferritin level**	Higher relapse rate for high-ferritin patients	AML-free survival 47%/73% Relapse 35%/16%	No significant associations
**Comment**	Ferritin and disease risk were independent risk factors	No difference in acute and chronic graft versus host disease (GVHD)	Biomarker risk groups defined by ferritin, CRP and albumin were associated with transplant-related mortality

**Table 7 ijms-26-05744-t007:** Effect of increased pretransplant ferritin levels on the clinical course after allogeneic stem cell transplantation [210,211,212,213,214,215,216,217]. The table presents the results for representative studies that included heterogeneous patients (at least 100 patients); all studies were retrospective and patients were transplanted during the time period 1996–2018. Arrows indicate increased/decreased parameter (↑↓).

Study Period References	Patients	Cut-Off	Overall Survival	Relapse Mortality	Nonrelapse Mortality	Acute GVHD	Chronic GVHD	Comment
1998–2005 [210]	309 patients, including AML, ALL, MDS, CML, CLL, lymphoma, myeloma and others.	400 μg/L	↓	↑	=	Nt	↓	Prognostic impact maintained in multivariate analysis, independent of CPR. Association between high ferritin and disease risk not examined.
1996–2006 [211]	264 patients including AML, CML, MDS, ALL, NHL.	600 μg/L	↓	=	↑	=	=	(i) Effects on survival and mortality maintained in multivariate analyses. (ii) Death from infections/organ failure more common with high ferritin. (iii) The frequency of patients with high CRP/low albumin was associated with high ferritin levels.
2000–2008 [212]	159 patients with AML, MDS and ALL.	1000 μg/L	↓	↑	↑	=	nt	Ferritin level and disease risk were independent risk factors; similar impact in standard- and high-risk patients.
1997–2005 [213]	590 patients with CML, AML, ALL, NHL.	2034 μg/L	↓	=	↑	Nt	nt	The adverse ferritin effect was seen especially for AML and MDS patients. Combination with albumin < 40 g/L did not influence the impact of ferritin on outcome, i.e., the effect seems independent of the acute-phase reaction.
2005–2006 [214]	190 patients with myeloid and lymphoproliferative malignancies.	1000 μg/L	↓	nt	↑	↑	nt	Only day +100 mortality was examined. Increased risk of blood-stream infections. The impact on survival seems stronger for myeloid malignancies.
2014–2018 [215]	290 patients with acute leukemia, MDS and lymphoma.	1500 μg/L	↓	↑	↑	=	=	All received matched sibling donor grafts. Increased non-relapse mortality was due to severe infections.
2000–2009 [216]	290 patients, mainly with AML/MDS but also myeloma, lymphoma, CML, CMN.	1358 μg/L	↓	↑	↑			High ferritin associated with reduced survival in all the periods of 0–6 months, 6–12 months, 1–2 years and 2–5 years; this was independent of erythrocyte transfusions and GVHD.
2004–2009 [217]	112 consecutive patients with hematological malignancies.	700 μg/L	↓	nt	↑	nt	nt	High ferritin associated with increased risk of sepsis/septic shock/organ failure. Diverse causes of death.

Abbreviations; ALL, acute lymphoblastic leukemia; CLL, chronic lymphocytic leukemia; CML, chronic myeloid leukemia; CMN, chronic myeloproliferative neoplasia; GVHD, graft versus host disease; MDS, myelodysplastic syndrome; NHL, non-Hodgkin’s lymphoma; nt, not tested.

**Table 8 ijms-26-05744-t008:** The three consecutive main characteristics of ferroptosis, description of important molecular and biological characteristics and their relevance in AML.

**1.** **Altered iron metabolism**
AML cells are at risk of developing or may have signs of cellular iron overload: (i) AML patients often show increased levels of soluble ferritin that can be taken up by the AML cells [157]. (ii) FTH and FTL are overexpressed compared with normal hematopoietic stem cells probably as a sign of increased iron load and the FTH expression then seems to be associated with NFκB activation leading to chemoresistance [248]. This is consistent with a ferritin-associated stabilization of both the general iron metabolism and the labile iron pool, finally leading to this resistance. (iii) Expression of the iron efflux protein ferroportin is decreased in AML cells compared with normal cells and this reduction may then predispose to iron retention and ROS formation [249]. (iv) The iron metabolism in AML cells is influenced by inflammatory activity in their microenvironment and increased levels of proinflammatory cytokines may then further contribute to decreased ferroportin levels [249].
**2.** **Production of reactive oxygen species (ROS)** [123,124]
High ROS levels in AML cells are associated with chemosensitivity [250], whereas low levels are associated with resistance [251]. The NOX family of oxidases [252] and the high mobility group box transcriptional protein [246] seem of particular importance for ROS production in AML cells. ROS modulates the function of signaling proteins through oxidation of cysteine residues and can thereby promote leukemogenesis through regulation of redox-sensitive transcription factors/enzymes/oncogenes and also promote genetic instability [253,254,255,256].
**3.** **Altered Lipid metabolism**
AML cells show increased levels of free fatty acids [123]. Mitochondria are the main source of ROS in AML [255] and mitochondrial metabolism is altered in AML cells compared with normal cells [256]. Certain mutations may also be associated with susceptibility to induction of ferroptosis, e.g., *IDH* [256] and *FLT3* mutations [257,258]. Lipid peroxide is cytotoxic and is regarded as a primary cause of ferroptosis [123,259,260].

## Data Availability

Not applicable.

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
