# Peer review of "Ferritin in Acute Myeloid Leukemia: Not Only a Marker of Inflammation and Iron Overload, but Also a Regulator of Cellular Iron Metabolism, Signaling and Communication"

_ijms, 2025, doi:10.3390/ijms26125744_

Round 1

Reviewer 1 Report

Comments and Suggestions for Authors

The manuscript offers a detailed and well-referenced review of ferritin’s diverse roles in acute myeloid leukemia (AML), extending beyond its traditional biomarker status to include regulatory functions in iron metabolism, immune modulation, angiogenesis, and leukemic cell survival. This review is a highly valuable, well-referenced, and detailed synthesis of current knowledge regarding ferritin in AML. It can serve as an excellent resource for both clinicians and researchers in hematology and cancer biology. 

It comprehensively covers both molecular mechanisms and clinical correlations, especially regarding prognosis and treatment outcomes. The structure is well-organized and guides the reader through molecular, cellular, and systemic levels of ferritin’s involvement in AML.

One drawback is that the manuscript is excessively long, with some redundant sections that could be streamlined for better clarity. Novel insights are limited, as much of the content recapitulates known data without offering new hypotheses or translational frameworks, and also, Therapeutic implications are underdeveloped. Lower are recommendations that would benefit the review:

Section 2 (Molecular Structure and Interactions of Ferritin):
Much of this section revisits structural biology already well-covered in prior reviews. Consider summarizing structural details more concisely and focusing on features most relevant to AML.

Section 3 (Microenvironmental Effects):
Subsections 3.2 through 3.5 contain overlapping discussions about microenvironmental interactions. These could be streamlined to reduce repetition and sharpen focus on clinically relevant mechanisms.

Section 4 (Acute Phase Response):
This section is very expanded and includes discussions on systemic inflammation, nutrition, and aging. A tighter focus on how these factors specifically relate to AML and ferritin would improve clarity and maintain relevance.

Overall, minor revision with suggested recommendations would benefit the quality of this paper.

Reviewer 2 Report

Comments and Suggestions for Authors

The presented manuscript, “Ferritin in Acute Myeloid Leukemia: not only a Marker of Inflammation and Iron Overload but also a Regulator of Cellular Iron Metabolism, Signaling and Communication”, describes the role of ferritin in Acute Myeloid Leukemia. To achieve a complete survey, the authors have included various experimental and clinical studies. They have extensively discussed the molecular structure and the molecular interactions of ferritin, the role of ferritin in influencing iron homeostasis in AML cells, and its role in the regulation of ferroptosis. 
The document is too long and difficult to read. It covers a huge amount of information from more than 330 references. I suggest that the authors reorganize the manuscript and shorten the explanations in the sections. This will make it more readable. In addition, some figures or diagrams could be included that visually present some of the results or the processes described. This will help readers to understand the manuscript more easily. 
Section 9 presents a number of therapeutic strategies, but as the authors emphasize, they have been studied mainly in experimental models. Can a critical assessment be made of which approaches are the most promising?
There is a technical problem with Figure 2 that needs to be corrected.

Author Response

See enclosed Cover Letter. 
